# Classification and phylogeny for the annotation of novel eukaryotic GNAT acetyltransferases

**Bojan Krtenic[1,2]\***, **Adrian Drazic[3]**, **Thomas Arnesen[1,3,4]**, **Nathalie Reuter[2,5]\***

1 Department of Biological Sciences, University of Bergen, Norway, 2 Computational Biology Unit, Department of Informatics, University of Bergen, Norway, 3 Department of Biomedicine, University of Bergen, Norway, 4 Department of Surgery, Haukeland University Hospital, Norway, 5 Department of Chemistry, University of Bergen, Norway

\* Bojan.Krtenic@uib.no (BK); Nathalie.Reuter@uib.no (NR)

**Data Availability Statement:** All relevant data are within the paper and its Supporting Information files.

**Funding:** NR acknowledges funding from the Research Council of Norway (#251247 and

## Abstract

The enzymes of the GCN5-related N-acetyltransferase (GNAT) superfamily count more than 870 000 members through all kingdoms of life and share the same structural fold. GNAT enzymes transfer an acyl moiety from acyl coenzyme A to a wide range of substrates including aminoglycosides, serotonin, glucosamine-6-phosphate, protein N-termini and lysine residues of histones and other proteins. The GNAT subtype of protein N-terminal acetyltransferases (NATs) alone targets a majority of all eukaryotic proteins stressing the omnipresence of the GNAT enzymes. Despite the highly conserved GNAT fold, sequence similarity is quite low between members of this superfamily even when substrates are similar. Furthermore, this superfamily is phylogenetically not well characterized. Thus functional annotation based on sequence similarity is unreliable and strongly hampered for thousands of GNAT members that remain biochemically uncharacterized. Here we used sequence similarity networks to map the sequence space and propose a new classification for eukaryotic GNAT acetyltransferases. Using the new classification, we built a phylogenetic tree, representing the entire GNAT acetyltransferase superfamily. Our results show that protein NATs have evolved more than once on the GNAT acetylation scaffold. We use our classification to predict the function of uncharacterized sequences and verify by *in vitro* protein assays that two fungal genes encode NAT enzymes targeting specific protein N-terminal sequences, showing that even slight changes on the GNAT fold can lead to change in substrate specificity. In addition to providing a new map of the relationship between eukaryotic acetyltransferases the classification proposed constitutes a tool to improve functional annotation of GNAT acetyltransferases.

## Author summary

Enzymes of the GCN5-related N-acetyltransferase (GNAT) superfamily transfer an acetyl group from one molecule to another. This reaction is called acetylation and is one of the most common reactions inside the cell. The GNAT superfamily counts more than 870 000

 

#288008). TA acknowledges funding from the Research Council of Norway (#249843), the Norwegian health authorities of Western Norway (#912176 and #F-12540) and the Norwegian Cancer Society (#PR-2009-0222). The funders had no role in study design, data collection and analysis, decision to publish, or preparation of the manuscript. URLs: * Norwegian health authorities of Western Norway: https://helse-vest.no/en/research-and-co-operation * The Research Council of Norway: https://www.forskningsradet.no/en/ * Norwegian Cancer Society: https://kreftforeningen.no/forskning/.

**Competing interests:** The authors have declared that no competing interests exist.

members through all kingdoms of life. Despite sharing the same fold the GNAT superfamily is very diverse in terms of amino acid sequence and substrates. The eight N-terminal acetyltransferases (NatA, NatB, etc. to NatH) are a GNAT subtype which acetylates the free amine group of polypeptide chains. This modification is called N-terminal acetylation and is one of the most abundant protein modifications in eukaryotic cells. This subtype is also characterized by a high sequence diversity even though they share the same substrate. In addition, the phylogeny of the superfamily is not characterized. This hampers functional annotation based on sequence similarity, and discovery of novel NATs. In this work we set out to solve the problem of the classification of eukaryotic GCN5-related acetyltransferases and report the first classification framework of the superfamily. This framework can be used as a tool for annotation of all GCN5-related acetyltransferases. As an example of what can be achieved we report in this paper the computational prediction and *in vitro* verification of the function of two previously uncharacterized N-terminal acetyltransferases. We also report the first acetyltransferase phylogenetic tree of the GCN5 superfamily. It indicates that N-terminal acetyltransferases do not constitute one homogeneous protein family, but that the ability to bind and acetylate protein N-termini had evolved more than once on the same acetylation scaffold. We also show that even small changes in key positions can lead to altered enzyme specificity.

## Introduction

Transfer of an acetyl group from one molecule to another is one of the most common reactions inside the cell. The rich and diverse, but structurally highly conserved, superfamily of GCN5-related acetyltransferases is one of the enzyme superfamilies able to catalyze the acetylation reaction [1–3]. Members of the GCN5-related acetyltransferase superfamily are able to accommodate numerous types of substrates including lysine sidechains [4–6] and N-termini of proteins [7], serotonin [8], glucosamine 6-phosphate [9], polyamines [10] and others. N-terminal acetylation is one of the most abundant protein modifications in eukaryotic cells, with over 80% of proteins susceptible to acetylation in multicellular eukaryotes [11]. The reaction entails transfer of an acetyl group from a substrate donor, most often acetyl coenzyme A, to a substrate acceptor, which is the N-terminus of the acetylated protein [12]. The abundance of N-terminal acetylation implies numerous effects of this modification on normal cell functioning and, indeed, it has been shown that N-terminal acetylation affects protein synthesis indirectly [13], protein folding [14, 15], protein half-life [16], protein-protein [17] and protein-lipid interactions [18], protein targeting [19], apoptosis [20, 21], cancer [22], a variety of congenital anomalies and autism spectrum disorder [23–26]. The importance of N-terminal acetylation is also striking in plants, where it is involved in plant defense and development [27], response to abiotic stressors like osmotic and high-salt stress [28] and response to other various types of biotic and abiotic stressors [29–32]. Despite the importance of N-terminal acetylation the number of N-terminal acetylating enzymes and cellular pathways remain unclear.

Thus far eight N-terminal acetyltransferases (NATs) have been discovered in eukaryotes with the last one identified in 2018 [33–48]. NATs are named NatA-NatH, by convention, and their catalytic subunits, which are the focus of this work, are named NAA10-NAA80. Each of the catalytic subunits has the same fold, called the GNAT fold. GNAT is the acetylation scaffold in the entire GCN5-related acetyltransferase superfamily [2, 3]. It is an α-β-α layered structure with a characteristic V-shaped splay between the two core parallel β-strands (usually β4 and β5 strands) (**Fig 1**). Together with the core strands, two loops (usually α1-α2 and β6-β7

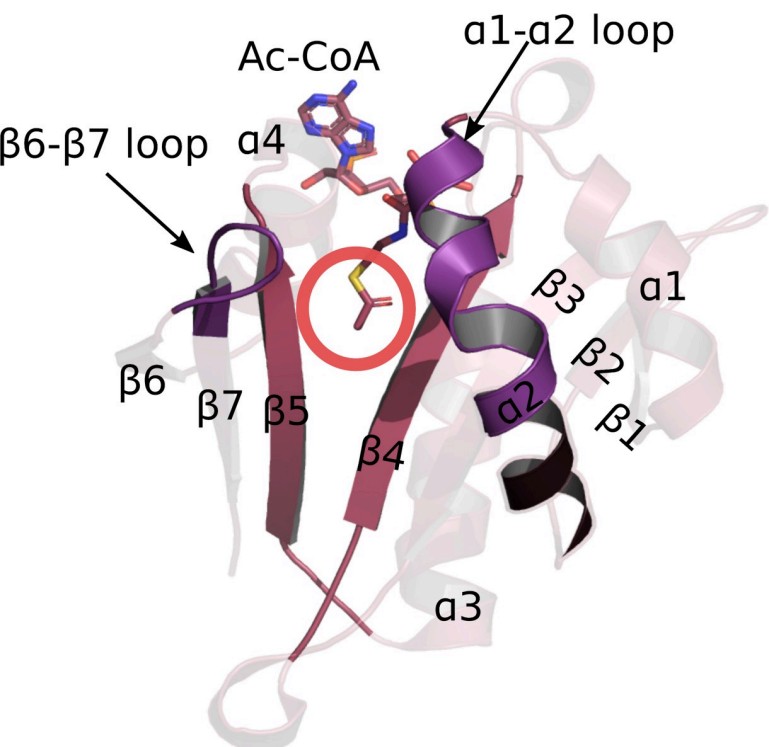

**Fig 1. GNAT fold is the acetylation scaffold in the acetyltransferase superfamily.** The fold positions the two substrates in such a way that the acetyl group of Ac-CoA approaches the N-terminus of the protein acceptor in the middle of the V-shaped splay between β4 and β5 strands–marked with the red circle. Four structural motifs have been identified in the GNAT fold: motif A consists of the β4 strand and α3 helix, motif B is the β5 strand and α4 helix, motif C includes the β1 strand and α1 helix, and motif D consists in the β2 and β3 strands [2].

loops) are involved in catalysis and substrate binding. They are located on one side of the splay. On the other side an α-helix (usually α3) common to all acetyltransferases binds Ac-CoA [2, 3] (**Fig 1**). While the β4 and β5 strands and the loops α1-α2 and β6-β7, are structurally quite conserved, their amino acid sequence varies with ligand specificity [2, 3, 49–56]. Consequently, the key determinants of the ligand specificity of an acetyltransferase are sequence motifs in the crucial positions on the GNAT fold.

Some NATs are more promiscuous than others when it comes to substrate specificity [7]. NatA is the most promiscuous NAT and it acetylates N-termini starting with A, S, T, C, V and G in fungi, plants and animals after the initial methionine is cleaved off [11, 12, 30]. NatD and NatH, on the other hand, have only one type of substrate. NatD acetylates N-terminal serine [35, 57], while NatH acetylates acidic N-terminus of processed actin [42, 56]. NatB, NatC, NatE, NatF and NatG have more relaxed specificity compared to NatD and NatH but are less promiscuous than NatA [58]. Usually the first two residues of a substrate protein determine whether the protein can be acetylated [11]. There is some overlap between *in vitro* specificities of NATs [59] and interestingly, it has been shown that some non-NAT acetyltransferases have the ability to N-terminally acetylate polypeptide chains. Glucosamine 6-phosphate acetyltransferases are one such example and were recently shown to *in vitro* acetylate N-terminal serine [60]. Besides Glucosamine 6-phosphate acetyltransferases, dual N-terminal activity has been demonstrated in plant plastids, where 8 GNAT acetyltransferases have been shown to be able to acetylate side chain lysines and protein N-termini [61]. The conservation of specificity from fungi to animals is high for some NATs but is not fully established for all eight identified

NATs. NatA and NatB, for example, have a quite well conserved specificity in all eukaryotes [11, 12, 30]. NatE has been shown to be catalytically inactive in yeast, unlike in multicellular eukaryotes where its specificity is well conserved [12, 17, 30, 53]. This has been shown only for yeast NAA50, however and it is unclear if all fungi NAA50s are catalytically inactive. NatC activity has been demonstrated in all eukaryotes [12, 30] with specificity conservation between yeast and human [36, 48] but plant NatC substrates still haven't been identified [62]. While some NATs are present in all eukaryotic kingdoms, some NATs are not; NatF is not present in fungi, for example [38]. NatG is located only in plastids of plant cells [41] and NatH has been identified only in animals [42]. NATs are referred to as a *family* of enzymes since they all acety-late the same type of substrate, namely protein N-termini, but the fact is that there is no deeper classification than at the *superfamily* level for all GNAT acetyltransferases.

The majority of all known types of acetyltransferases are members of the same Pfam [63] family (Acetyltransf_1, code: PF00583) which contains almost 50% of the entire acetyltransfer-ase clan (Pfam code: CL0257). The Acetyltransf_1 Pfam family contains 120,379 sequences out of the 280,421 sequences of the acetyltransferase clan and consists of numerous types of acetyl-transferases. PROSITE [64] does not differentiate between different types of acetyltransferases either and recognizes four types of GNAT fold: GNAT (PS51186), GNAT_ATAT (PS51730), GNAT_NAGS (PS51731) and GNAT_YJDJ (PS51729). The CATH database [65] offers a slightly better classification than Pfam or PROSITE, but CATH does not accurately differenti-ate between all known NAT sequences. As a result, and despite extensive efforts on the experi-mental front, the current classification of acetyltransferases is based on a collection of ligand specificity assays which can only sparsely cover the variety of enzymes in the superfamily.

Several studies have identified a large number of proteins that can be N-terminally acety-lated [33, 39, 66–69]. Much of the identified acetylated N-termini can be explained by cur-rently known NATs [11, 66]. However, we do not know whether or not known NATs acetylate other exotic N-termini found to be N-terminally acetylated in cells, such as those with acety-lated initial tyrosine (PCD23_HUMAN, KS6A5_HUMAN, etc) [66]. N-terminal acetylation events following post-translational protease action are not well characterized either; known NATs except NatF, NatG and NatH sit on the ribosome and catalyze cotranslational acetyla-tion [58]. Therefore, there might be unidentified NATs in eukaryotes responsible for such events. The lack of a classification of acetyltransferases at the *family* level hinders functional annotation based on sequence similarity, and hence slows down the identification of new NATs.

In order to create a better classification framework for the eukaryotic acetyltransferase superfamily we used a combination of bioinformatics sequence analysis consisting in sequence similarity networks (SSNs), motif discovery and phylogenetic analysis. We showed that N-ter-minal acetyltransferases do not constitute one homogeneous *family*, even though they acetylate the same type of substrate. Our analyses all converge to the conclusion that NATs evolved more than once. Finally, we could predict and experimentally verify that two uncharacterized sequences from fungi closely related to two known NATs, NAA50 and NAA60, encode NAT enzymes targeting specific protein N-terminal sequences. This experimental validation gives us confidence that our classification will be a valuable tool for identification and annotation of new superfamily members.

## Results and discussion

### Sequence similarity networks (SSNs)

We collected from UniProt all eukaryotic sequences matching the GNAT signature defined by PROSITE. The collected sequences were then filtered at 70% identity to reduce the size of the

dataset, using h-cd-hit [70], which resulted in a dataset of 14,396 sequences. We also collected a second dataset restricted to the sequence of the GNAT-domains. We generated SSNs for each of the datasets using EFI-EST [71]. By adjusting the E-value and alignment score threshold for drawing SSN edges (S1A Text), we created an SSN with the highest probability of having isofunctional clusters. Both SSNs resulted in a sparse topology indicating a high sequence diversity in the acetyltransferase superfamily (S1B Text). The convergence ratio of the SSN built from the full-length sequences is 0,008 and it is equal to 0,009 for the network build from the GNAT domains only. This illustrates the high level of divergence between acetyltransferases.

We used the clusterONE algorithm [72] through Cytoscape [73] to determine the boundaries between each cluster in our SSNs. We identified 232 clusters in the full-length sequence SSN and 221 clusters in the GNAT domain SSN. Since the results for both networks are highly similar, we opted to use the full-sequence SSN for further analyses. When applying clusterONE to SSNs, we used the percentage of sequence identity as edge weight to make sure that clusters are identified based on a reliable measure of similarity. Dense regions thus correspond to very similar sequences. We also observe that, with few exceptions, known acetyltransferases of one particular function never appear in multiple clusters. We can thus reasonably assume that the clusters in our SSN are isofunctional.

In order to better visualize the relationships between clusters we represented the SSNs as simplified, "pivot", networks. Each cluster of the original SSN is represented by a single node. An edge between nodes in the simplified network is drawn where there was at least one edge between any nodes of the two corresponding clusters in the original SSN (**Fig 2** and S1C Text). The main topological characteristics of the SSNs are network sparsity, the resulting absence of SSN hubs, several connected components that contain a varying number of clusters, and a large number of isolated clusters (**Fig 2**). We identified 48 clusters with known acetyltransferases and 184 completely uncharacterized clusters. The majority of proteins in our SSN are from fungi (S1C Text), but all eukaryotic kingdoms are represented. There is a total of 80 *Homo sapiens* proteins in the SSN, spread into 21 clusters. The observed clustering is not based on taxonomy, but instead correlates with ligand specificity (S2 Text). Interestingly, acetyltransferases that acetylate the same type of substrate (e.g. either N-termini of proteins or histones) are not necessarily found within the same connected component but are scattered over the SSN. This is the case with NATs, which are found clustering together with other types of acetyltransferases rather than forming one homogeneous group. This is the first indicator that NATs do not constitute one homogeneous family but have, rather, evolved more than once on the same scaffold.

## Identification of five NAT groups: sequence motif fingerprints and structure comparison

**Sequence fingerprints.** Known NATs do not all inhabit the same connected components of the SSN (**Fig 2**), which indicates NATs are not one homogeneous family of acetyltransferases. We used MEME [74] from the MEME suite [75] to identify motifs in each of the SSN clusters. Sequence motifs of highly conserved residues were detectable for each of the clusters. Based on the similarity between motifs and on the clustering of the NATs in the SSN, we defined five different groups of NATs (**Fig 2**). We subsequently calculated sequence motifs for each of these groups. The motifs are shown in **Fig 3** and **Table 1**.

Group 1 consists of Group 1a with NAA10 and NAA20, and Group 1b that contains NAA30 of all eukaryotic kingdoms. NAA10 and NAA20 are in the same connected component, while NAA30 is found in a single isolated cluster. Sequence motifs that are important for

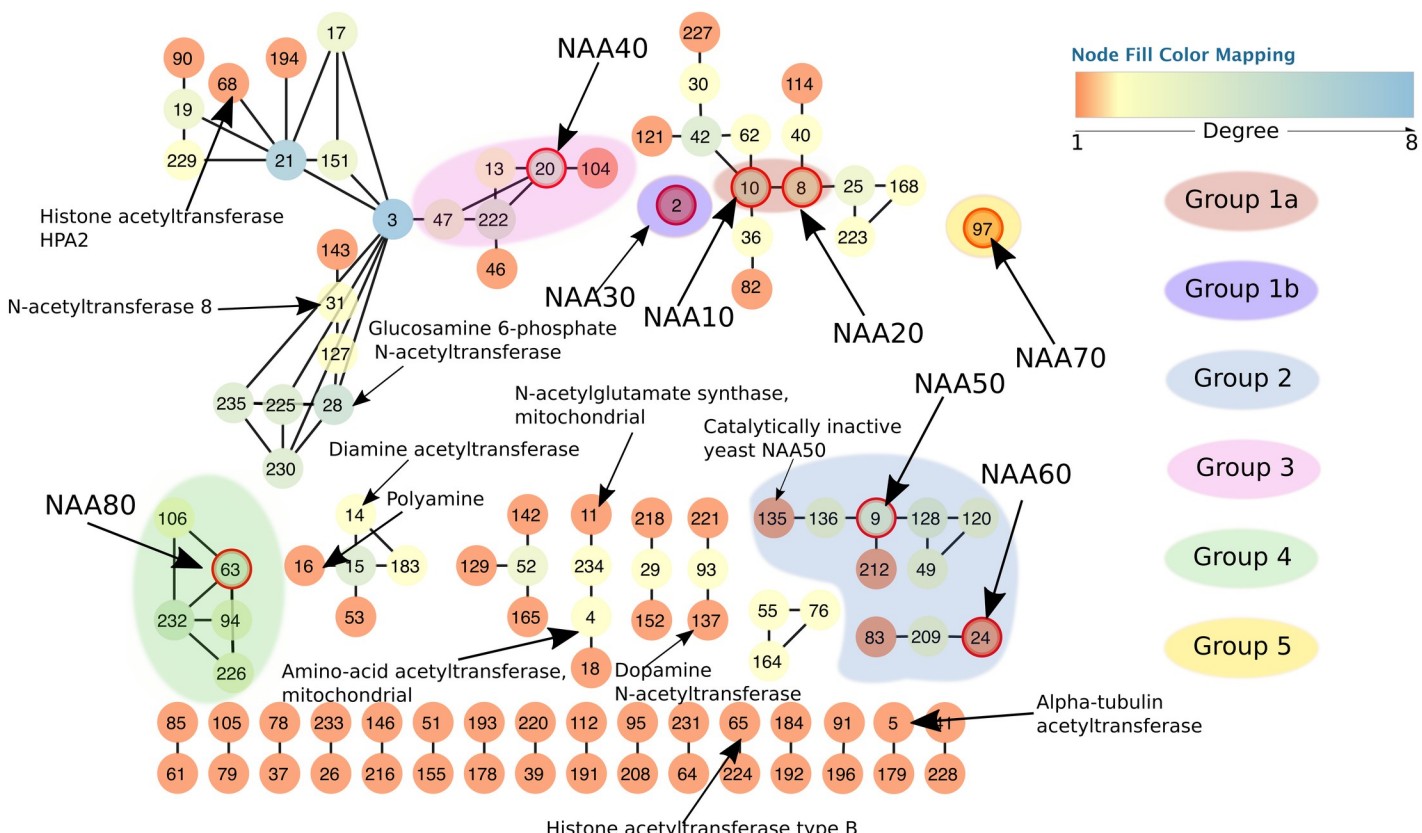

**Fig 2. Simplified view of the resulting sequence similarity network.** Each node represents one cluster from the original network. Edges connect two nodes in the simplified network if there is at least one edge between any nodes of the corresponding clusters in the full network. Node colors correspond to their degree, i.e. the number of connections to the neighboring nodes. Each node in the network has a unique number assigned by clusterONE [72]. The numbers serve as cluster names in cases where the cluster is uncharacterized. All nodes circled in red are known and experimentally confirmed N-terminal acetyltransferases (10 –NAA10, 8 –NAA20, 2 – NAA30, 20 –NAA40, 9 –NAA50, 24 –NAA60, 97 –NAA70 and 63 –NAA80). Of importance is also cluster 135, which contains the catalytically inactive yeast NAA50. The network shows four NAT groups. Group 1 consists of two subgroups–Group 1a which contains NAA10 and NAA20 and Group 1b which contains NAA30.

binding of substrate and acetylation in both subgroups of Group 1 NATs are localized on the α1-α2 loop, the β4 and β5 strands and the β6-β7 loop [51, 52, 54] (S3A Text) and this is true for groups 2 and 3 as well. Group 2 consists of NAA50 and NAA60. NAA50 and NAA60 do not cluster together in the SSN but the resemblance between their key sequence motifs (S3B Text) justifies placing them in the same group. Group 2 also contains a catalytically inactive yeast NAA50, a member of the fungal *Sacharomicetaceae* family (S3C Text). This inactive NAA50 forms a separate cluster, numbered 135, in the SSN (**Fig 2**). Sequence motifs on key secondary structure elements are not strictly conserved between cluster 135 and the catalytically active NAA50 (cluster 9). This explains why the inactive enzyme clusters separately from cluster 9. It is important to mention that NAA60 of animals and plants do not share the same cluster (S2C Text). Plant NAA60 bears some slight, but potentially important differences on the α1-α2 loop, where a negatively charged E in animals is replaced by a positively charged K in plants. The key tyrosine of β6-β7 loop is still present in plants, but unlike animal NAA60 which has thee tyrosines in this loop, plant NAA60 only has one. We define Group 3 around NAA40. Some plant NAA40 form a separate cluster and some others share the same cluster as animal NAA40 sequences. A striking characteristic of NAA40 is its long α0 helix and the position of its α1-α2 loop [55] which extends over and covers the binding site where the β6-β7 loop lies in other NATs. Group 4 is defined around NAA80 which is structurally different

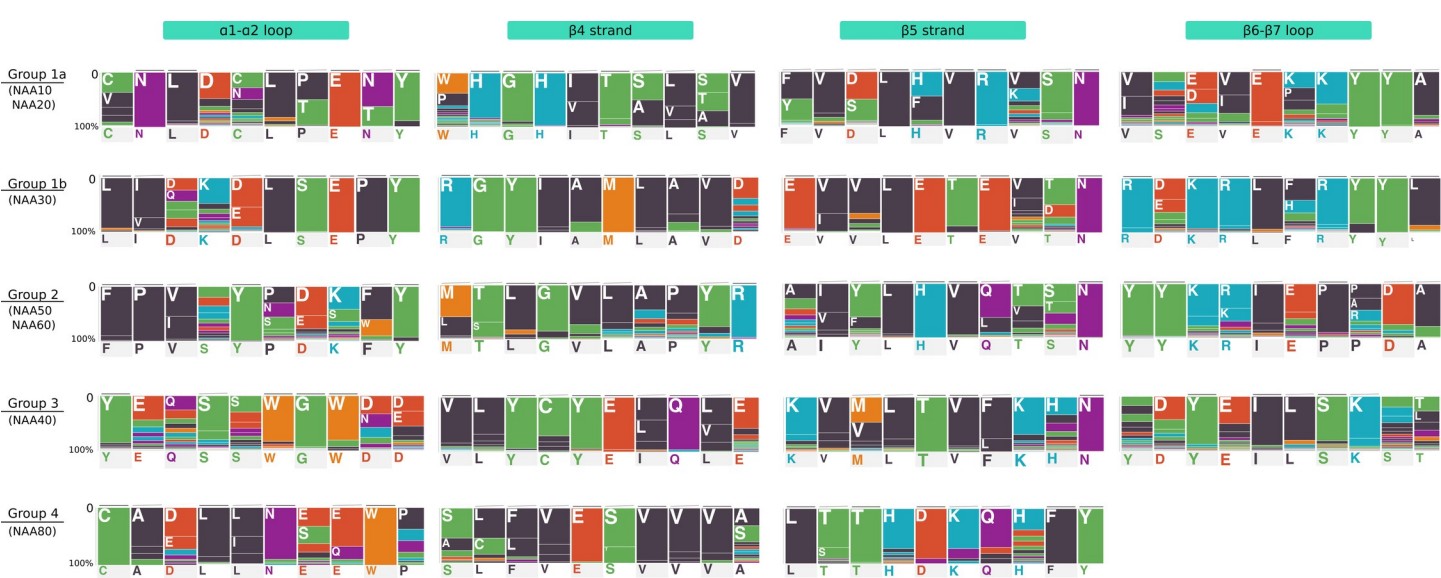

**Fig 3. Characteristic sequence motif fingerprints of NAT Groups 1 to 4.** Sequence motifs were calculated as described in the Material and Methods section and using sequences from the SSN clusters. Each position in the motif is represented by a colored bar and a one-letter code for the amino acid frequently found at that position in the GNAT fold. The height of colored bar is proportional to the frequency of the corresponding amino acid. The colors correspond to the type of amino acid (blue: R, H, K; red: D, E; green: C, S, G, Y, T; black: P, F, V, L, I, A; orange: M, W; purple: N, Q). Group 5 is not shown as there is no structure of NAA70 available.

from the first three groups. Its surface shows a large cleft which is covered by loops in all other NAT structures available to date [56]. The need for a larger ligand binding site is explained by the fact that NAA80 has evolved to catalyze N-terminal acetylation of fully folded actin and harbors an extensive binding surface to actin [76]. Finally, Group 5 contains NAA70 which is a chloroplast NAT discovered in *Arabidopsis thaliana* [41]. NAA70 is closer to bacterial acetyl-transferases than to the eukaryotic ones in Groups 1 to 4. A BLAST search against the NCBI non-redundant database [77] and excluding green plants, suggests that NAA70 is most similar to cyanobacterial proteins with the best hit being a protein from *Gleocapsa sp* (29,7%id over 62% query cover). We also found that NAA70 shares a high percentage of sequence identity with *Enterococcus faecalis* acetyltransferase whose structure has been solved (PDB code 1U6M). Unfortunately, there is not enough reliable structure information on NAA70 to be able to map the position of the key sequence motifs onto the secondary structure elements.

While there are important differences between each of the groups in terms of sequence motifs, some similarities emerge (**Fig 3**). They are especially obvious between groups 1a, 1b and 2, where we can observe a well conserved tyrosine in the α1-α2 loop (**Fig 3** and S3D Text) and most importantly, another conserved tyrosine in the β6-β7 loop (**Fig 3** and S3D Text). This tyrosine is essential for function and is strictly conserved in all members of groups 1 and 2 [49–51, 54, 78] with the exception of the catalytically inactive fungal Sacchormycetaceae

**Table 1. Regular expressions for key sequence motifs of NATs Groups 1 to 4.** All regular expressions were calculated using MEME from MEME Suite [75].

| Group / ss element | α1-α2 | β4 | β5 | β6-β7 |
|---|---|---|---|---|
| **Group 1a** | [CV]NLD[CN]L[PT]E[NT]Y | [WP]HGH[IV]T[SA][LV][STA]V | [FY]V[DS]L[FH]VR[VK]SN | [VI]X[ED][VI]E[KP]KYYA |
| **Group 1b** | L[IV][DQ]K[DE]LSEPY | RGYIAMLAVD | E[VI]VLETE[VI][TD]N | R[DE]KRL[FH]RYYL |
| **Group 2** | FP[VI]XY[PNS][DE][KS][FW]Y | LYI[ML][TS]LGVLAPYR | A[IV][YF]LHV[QL][TV][ST]N | HS[FY]LPYYYSI |
| **Group 3** | YEQSSWGW[DN][DE] | VLYCYE[IL]Q[LV]E | KV[MV]LTV[FL]KHN | |
| **Group 4** | CA[DE]L[LI]N[ES][EQ]W[PK] | [SA][LC][FL]VE[ST]VVV[AS] | L[TS]THDKQHFY | |

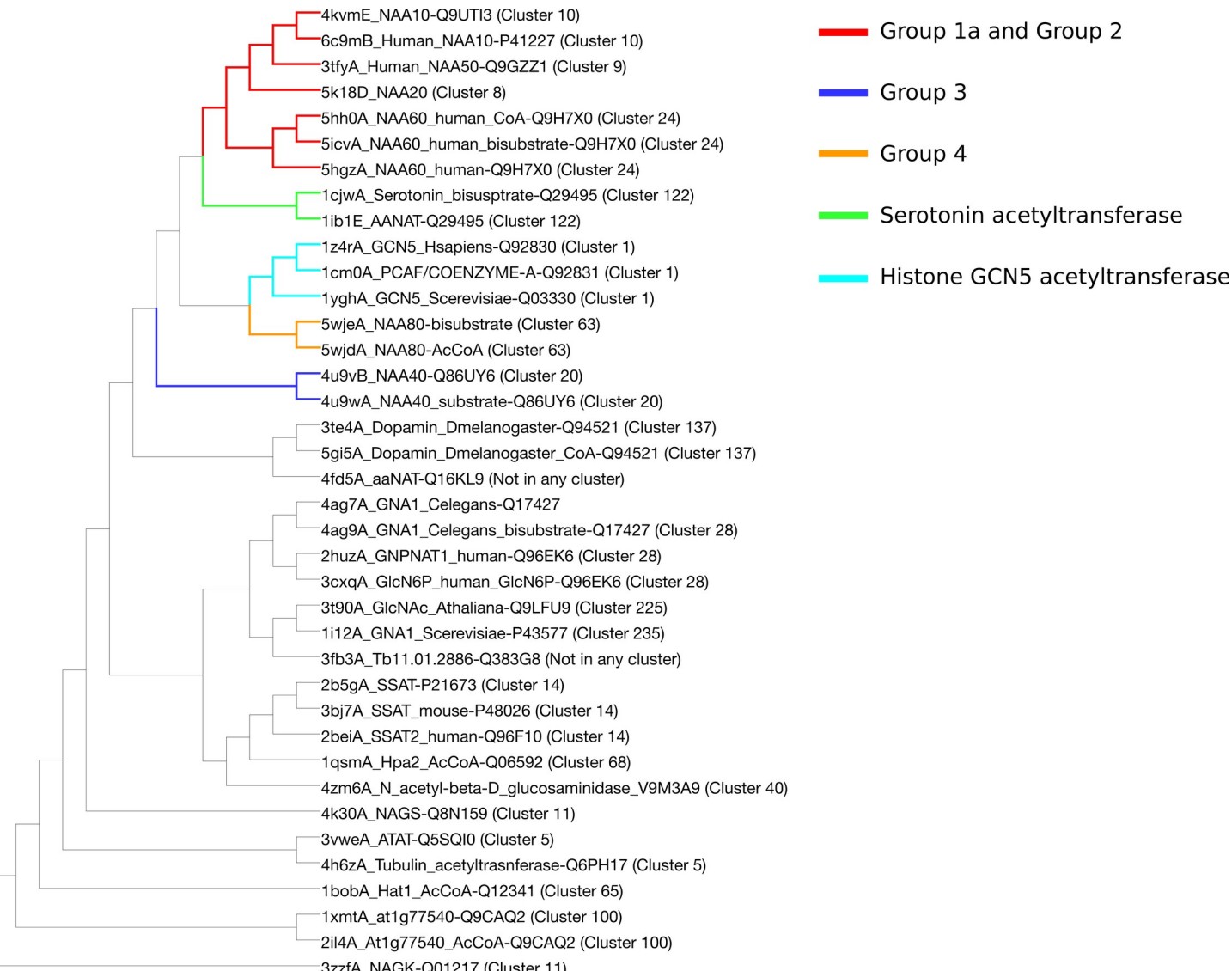

**Fig 4. DALI dendrogram for structural similarity between acetyltransferases.** The dendrogram is the result of hierarchical clustering of structures. The known NATs are closer to one another than to the rest of the superfamily. Note the non-NAT acetyltransferases located close to known NATs.

NAA50 where this tyrosine is lost [53]. The tyrosine in the α1-α2 loop is conserved in all NATs of Group 1a, Group 1b and Group 2 [49, 50, 79] except for NAA20 where it can be replaced by phenylalanine [51]. Group 3 and group 4 motifs clearly differ from those of Group 1 (**Fig 3**). Compared to the other groups, strands β4 and β5 stand out in groups 3 and 4 where they play a major role in substrate binding and catalysis. Interestingly their sequence motifs and key residue positions differ between the two groups [55, 56].

**Structure comparison.** We compared structures of acetyltransferases to one another using the DALI server [80]. DALI computes and compares intramolecular distance from each structure, and no sequence information is used. Our dataset consists of structures of 38 catalytic subunits of acetyltransferases, all belonging to our SSN. The resulting dendrogram (**Fig 4**) illustrates the degree of similarity between all structures in the dataset. It shows a classification that overlaps with that of the SSN. In addition, it highlights that Group 1a and Group 2 are

more similar to one another than to the other NATs, and to the rest of the entire superfamily. Structure of NAA30, around which we built Group 1b, has not been solved, but based on the sequence similarity it is safe to assume that it would be found in the same part of the dendrogram as Group 1a and Group 2. NAA40 (Group 3) is the NAT closest to NAA80 (Group 4), to the histone acetyltransferase GCN5, to the dopamine N-acetyltransferase and to the arylalkylamine N-acetyltransferase. The proximity of NAA40 and NAA80 is only observed in the structure-based classification and was not observed in the sequence similarity network, although NAA40 and NAA80 are closer to one another than to other known NATs in the SSN. NAA80 is also close to the histone acetyltransferase GCN5. Groups 1a, 2, 3 and 4 of NATs are more similar to one another than to the rest of the superfamily when structures are compared, but this is not the case when sequences are compared. In-between these 4 groups one finds non-NAT acetyltransferases, namely the histone acetyltransferase GCN5 (cluster 1) and serotonin acetyltransferases (cluster 122).

It is important to mention that while we can see differences in structures of different acetyltransferases, they are still quite similar to one another. We verified the proximity of the structures by building a network based on a structure similarity matrix. The resulting network is random with all nodes connected to all nodes when we use a Z-score higher than 2, which is considered to be significant as a threshold for an edge between two nodes [81]. The Z-score threshold needs to be increased to at least 15 for cluster separation in the similarity network to appear (S1 Fig).

**Similarities and differences between groups.** Groups 1 and 2 are highly similar. Because of the similarities between NAA10 in one hand, and NAA30 in the other hand, we placed them in the same group of NATs but defined two subgroups (1a and 1b), as the NAA30 cluster is not connected to those of NAA10 and NAA20. Sequence motifs on the α1-α2 loop, β4 and β5 strands and β6-β7 loop are characteristic to both subgroups and represent its signature. Protein N-termini starting with a small residue, which is exposed after removal of the initial methionine, and protein termini with the initial methionine can be acetylated by this group of NATs [7]. Even though these enzymes are obviously closely related, they employ different solutions to bind and acetylate substrates. Slight changes are sufficient to shift the substrate specificity of the GNAT fold. The same GNAT elements in Group 2 of NATs, which contains NAA50 and NAA60, are important for substrate binding and catalysis [49, 50]. Group 2 NATs acetylate protein N-termini starting with a methionine [49, 50]. Interestingly, Group 2 of NATs contains an inactive yeast NAA50 (found in cluster 135), which does not contain the characteristic β6-β7 tyrosine involved in substrate binding and which is present in Group 1 and Group 2 NATs [53]. The inactive yeast NAA50 does not have the Ac-CoA binding motif either [53]. It is therefore expected that its function is not the same as the function of NAA50 sequences which have all substrate interaction sites conserved. By comparing sequences and available structures we found that these important binding sequence motifs are absent only in the Saccharomycetaceae from the fungal Ascomycota phylum. Other families from this phylum and members of other fungal phyla seem to have conserved substrate binding sequence motifs when compared to NAA50 of other eukaryotic kingdoms. Large differences between Group 1a and Group 2 exist in the way the substrate binds to the enzyme and also in the position of the catalytic residue on the fold. The difference in catalytic strategy between Group 1a and Group 2 enzymes can be illustrated by drawing a horizontal line through the middle of the V-shaped splay across the β4 and β5 strands (Cf Fig 1); in Group 1a the active site would be above the line, while it would be below the line in Group 2. Interestingly, catalytic residues of Group 2 are conserved in Group 1a, but they are not catalytically active in Group 1a [54]. The two groups share two conserved tyrosines in each of the β6-β7 and α1-α2 loop. Both are involved in substrate binding. In Group 1a of NATs, residues at positions upstream of the

mentioned tyrosine (positions -2 and -4 with respect to the Tyr) are also involved in substrate binding, but the same positions in Group 2 do not seem to be as important for binding the substrate, even though position -4 shows mutational sensitivity [51].

There are important differences between groups 3 and 4, and each is very different from the other groups. Group 3 contains NAA40, the NAT with the specificity towards the histone H4 and H2A N-termini (sequence: SGRG) [55]. The first major difference between NAA40 and Group 1 and 2 NATs is the fact that the β6-β7 loop in NAA40 has no role in determining substrate specificity [55] and does not carry an invariable tyrosine. While the α1-α2 loop of NAA40 plays a role in substrate binding [55], just like in groups 1 and 2 NATs, it has a different position in the 3D structure and its sequence motif does not bear any resemblance with the conserved motifs of groups 1 and 2. Our SSNs and phylogenetic tree all show a large distance between Group 3 and Groups 1 and 2.

Group 4 is defined by NAA80, the most recently discovered N-terminal acetyltransferase [42]. The β6-β7 loop of NAA80 is not conserved and does not play an important role in acetylation. As in most other NATs the α1-α2 loop plays an important role in substrate binding and is well conserved [56]. The α1-α2 loop sequence motif is different from those of other NATs. Moreover, NAA80 has a wider substrate binding groove between the α1-α2 and β6-β7 loops. This structural feature supports classifying NAA80 into a different NAT type. In addition NAA80 does not have the α1-α2 and β6-β7 tyrosines found in Groups 1 and 2. Our results confirm, over a larger set of sequences, an observation that has been reported earlier [56].

## Phylogeny

We used the clustering information obtained from the smallworld SSN (S4A Text) to generate the dataset for phylogeny. We selected 3 random sequences per SSN cluster and created an MSA for the structural motifs A (β4 and α3) and B (β5 and α4) of the GNAT fold (See **Fig 1** and S5A Text). They are the most conserved structural motifs across the superfamily [2] and their alignment yields a better MSA than a whole-sequence alignment would. Note that NAA70 was not included in the MSA because it is not found in the connected component of the SSN used to generate the phylogeny dataset. The constructed tree is not rooted because of the lack of a good outgroup and it does not inform on the direction of evolution within the superfamily. It therefore represents a network of similarity of acetyltransferases which reflects well the classification of NATs presented above and obtained from the SSN, the analyses of the sequence motifs and evidence from the literature.

The phylogenetic tree is shown in **Fig 5**. The branching in the tree yields five categories of NATs, without taking into account the group built around NAA70 which is not represented in the tree. These five categories correspond to the Groups 1a, 1b, 2, 3 and 4 obtained based on the SSN and motif similarity (Cf **Fig 2**). NATs from Group 1a (NAA10, NAA20), Group 1b (NAA30) and Group 2 (NAA50 and NAA60) are closely related according to the tree (**Fig 5**) and according to the smallworld SSN (S4A Text). This is in agreement with evidence that NAA10 and NAA50 have evolved from the same archaeal ancestor [82]. An archaeal N-terminal acetyltransferase, whose structure was solved by Liszczak and Marmorstein [82], can acetylate substrates of both NAA10 and NAA50. The archaeal enzyme employs catalytic strategies from both of these enzymes. It is most likely, as the authors suggested, that NAA10 and NAA50 evolved from this common ancestor. NAA10 and NAA20 do not share the same branch with NAA30 despite the similarity between their sequence motifs on key secondary structure elements of the GNAT fold. Groups 3 and 4 appear close to each other; NAA40 and NAA80 are close to one another. Several distinct branches of the tree carry a particular type of acetyltransferases (**Fig 5**), but even within some of these branches we see acetyltransferases

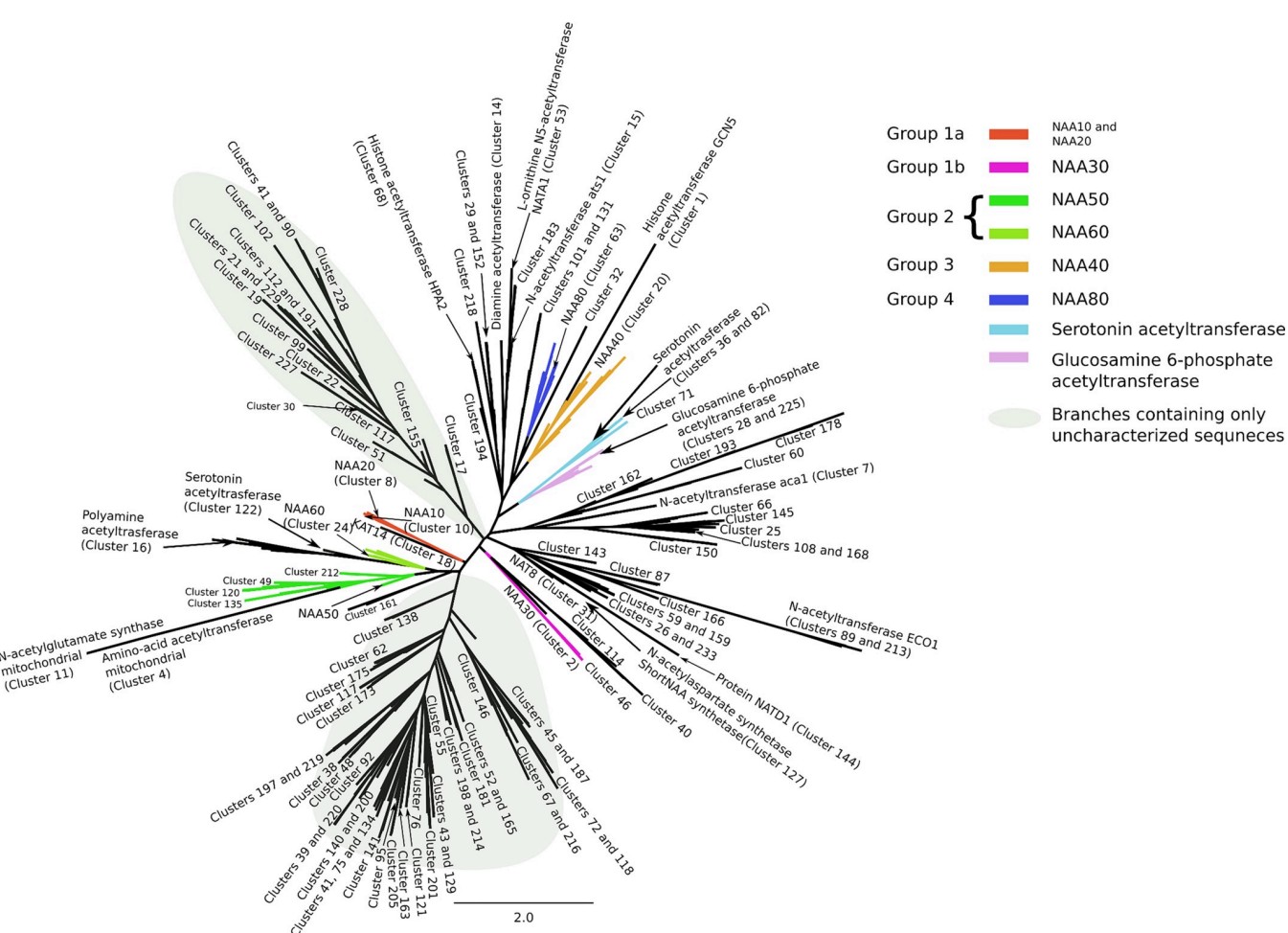

**Fig 5. Unrooted phylogenetic tree of the acetyltransferase superfamily.** The tree contains only those sequences for which we could find significant relationships in the SSN. A gray background is used to highlight the branches on the tree that are populated exclusively by uncharacterized sequences.

acetylating different types of substrates. We have mapped the SSN clusters to the tree in order to observe relationships between NATs and other identified acetyltransferases.

The tree shows several acetyltransferases, annotated as non-NAT enzymes, close to the Group 1a NATs (red branches). For example, a histone acetyltransferase (KAT14 –cluster 18) is found close to Group 1a of NATs (NAA10 and NAA20) and these sequences are the closest relatives according to the tree. An MSA of these acetyltransferases (S5B Text) reveals that KAT14 and sequences in Group 1a share sequence motifs. Indeed, the best conserved sequence motif found in Groups 1 and 2, located on the β6-β7 loop, is conserved in KAT14, as well. The β6-β7 loop motif contains a tyrosine present in all Group 1 and Group 2 N-terminal acetyl-transferases (NAA10, NAA20, NAA30, NAA50 and NAA60), with the exception of the fungal Saccharomycetaceae family. The same tyrosine is present in the KAT14 β6-β7 loop (S5B Text). This tyrosine has been shown to be essential for substrate binding [49, 54, 78] and it has been suggested that the size and flexibility of the β6-β7 loop plays an important role in substrate rec-ognition [2, 83]. Based on similarity between the β6-β7 loop of KAT14 and the NATs from Groups 1 and 2 and given the fact that the β6-β7 loop differs in size and primary sequence in other acetyltransferases, it is not excluded that KAT14 might be able to accommodate the same type of substrate as NATs and acetylate N-termini of proteins.

Looking now more specifically at the branches around Group 2 (green branches), we can see that clusters 49, 120, 128, 135, 136 and 212 are found close to NAA50 (cluster 9) (**Fig 5**). Clusters 83 and 209 are found close to NAA60 in the phylogenetic tree as well (**Fig 5**). Additionally, according to the tree, clusters 122, 78, 16 and 37 are closely related to NAA60. Cluster 122 is a serotonin N-acetyltransferase [84] and forms a single cluster in the stringent SSN. There are similarities between serotonin N-acetyltransferase and NAA60. Like NAA60, serotonin acetyltransferase has a long β3-β4 loop, unlike other NATs [50] (S5C Text). Catalytic residues are positioned similarly in both enzymes. Tyr97 in NAA60 and His 120 in serotonin acetyltransferase have equivalent positions in on the GNAT fold (S5C Text). The other catalytic residue of NAA60 (His138) and cluster 122 serotonin acetyltransferase (His122) are both located in the core of the GNAT fold (S5C Text) even if their positions are not equivalent. Cluster 16 is annotated as a polyamine acetyltransferase [85, 86] and it establishes weak connections with, among few others, cluster 14 (diamine acetyltransferases) in the stringent SSN.

Cluster 161 is close to NAA50, NAA60 and their surrounding clusters and the MSA of NAA50, NAA60 and sequences in cluster 161 shows many conserved key residues (S5D Text). Cluster 161 contains only sequences of *Caenorhabditis tropicalis* and is highly similar to both NAA50 and NAA60. It might therefore acetylate substrates similar to those acetylated by NAA50 and NAA60.

In Group 3 clusters 104, 222, 13 and 47 are close to NAA40 (ochre branches) (**Fig 5**). Additionally, clusters 111 and 176 are close to NAA40 and surrounding clusters (**Fig 5**). Sequences in cluster 176 are annotated as NAA40. It is unclear whether cluster 111 is also a NAA40 or if it has a different substrate specificity.

In Group 4 of NATs (dark blue branches), clusters 94, 106, 226 and 232 are close to NAA80 (cluster 63) (**Fig 5**), This group of sequences is close to cluster 32. Another branch, branching from the NAA80 branch, contains clusters 14 (Diamine acetyltransferases), 15 and 53 (Tyramine N-feruloyl transferase 4/11). In addition, on the same branch, but closer to clusters 14, 15 and 53 than to NAA80, lie uncharacterized clusters 29, 152 and 218. Clusters 29, 152 and 218 are closely related, according to our tree, with cluster 68 (Histone acetyltransferase HPA2 [87]) and 194.

Histone acetyltransferase GCN5 (cluster 1) is found on the same branch as NAA40 on the phylogenetic tree and, also, close to NAA40 and NAA80 on the structure similarity dendrogram (**Fig 4**). The MSA between NAA40 and acetyltransferases from cluster 1 shows some conservation between these two types of acetyltransferases, but none of the functional key residues for NAA40 are conserved in sequences from cluster 1 (S5E Text). Judging by the branching of our tree, NAA40 and NAA80 are closer to one another than to other NATs (**Fig 5**). Indeed, these two NATs do not share any of the characteristics of Group 1 and Group 2 NATs. Their separate branching is in agreement with the assumptions we made about N-terminal acetyltransferases evolving more than once, which was based on the topology of our SSNs and on the sequence motif composition.

Eight enzymes in *A. thaliana* mitochondria and chloroplasts have been shown to be able to acetylate protein N-termini [61]. Three out of those eight enzymes are present in our phylogenetic tree (cyan branches) (**Fig 5**). Clusters 36, 71 and 82 are closely related according to our tree, which is in agreement with observations made by Bienvenut and colleagues in their study [61], where they group these three enzymes into the same subgroup. Interestingly, clusters 36, 71 and 82 are found branching close to Glucosamine -6-phosphate acetyltransferase (**Fig 5**), which has also been shown to be able to acetylate protein N-termini [60]. The remaining five *A. thaliana* plastid enzymes with demonstrated N-terminal acetylation activity are found in clusters 50, 73, 74, 88 and 97 but they are not present in the tree; their inclusion lowers the robustness of the phylogenetic tree because of their high dissimilarity to the rest of the superfamily.

### Prediction of new acetyltransferases

**Predictions based on SSN and sequence motifs.** The initial SSN (**Fig 2**) revealed clusters containing uncharacterized sequences around clusters of known NATs. In what follows, we investigate how similar or different from known NATs these uncharacterized clusters are. To that goal, we checked for the presence in these sequences of the motifs we had characterized for the groups of known NATs (Table 1). Proteins belonging to those clusters and displaying sequence motifs close to the NAT motifs are likely to be NATs.

Around Groups 1a and 1b, no neighboring cluster showed sequence motifs close to those found in clusters 2 (NAA30), 8 (NAA20) and 10 (NAA10). Therefore, none of the connected clusters to either NAA10 or NAA20 were considered as potential new NATs. In group 2, the connected components around clusters 9 (NAA50) and 24 (NAA60) contain uncharacterized clusters numbered 49, 120, 128, 135, 136, 212 and 83 and 209, respectively (Cf Fig 3). All clusters, in addition to an isolated cluster numbered 207, have sequence motifs resembling that of group 2 (S6A Text) but also displaying significant differences. In the 6 clusters around cluster 9 (NAA50) (**Fig 2**), NAA50 is the only confirmed acetyltransferase and we found sequences annotated as NAA50 both in clusters 9 and 135. There are X-ray structures available for both of these clusters, and proteomics and biochemical data show that they may differ in their substrate specificity [53, 88]. We observe a difference in sequence motifs; a mutation-sensitive phenylalanine [49] in the α1-α2 loop of NAA50 (first Phe in the motif of group 2 shown on Fig 3) is replaced by a less bulky leucine in sequences from cluster 135 (**Fig 6A and 6C**). We observe the same differences in the α1-α2 loop between cluster 9 (NAA50) and uncharacterized clusters 49, 120, 128 and 212 (**Fig 6A**). Two residues downstream from the leucine/phenylalanine substitution, we observe a conserved isoleucine in cluster 120 instead of a highly conserved valine in NAA50. This valine forms van der Waals contacts with the substrate in NAA50 [49] and is, thus, important for substrate binding. Moreover, the α1-α2 loop in cluster 120 sequences contains two conserved prolines (**Fig 6A**) unlike NAA50 that contains only one (**Fig 3**). The characteristic β6-β7 motif of NAA50 (**Cf Table 1 and Fig 3**) is not present in cluster 135, which doesn't have the conserved tyrosine in this loop (second tyrosine of the sequence motif on Fig 3). Structurally, the differences between cluster 9 (NAA50) and cluster 135 enzymes are precisely in the β6-β7 loop, which is longer in cluster 135 structure (S6B Text). Sequences from all other clusters around cluster 9, carry the same β6-β7 loop motif as NAA50 (**Fig 6A**). Finally, there are differences in sequence motifs carried by the β4 strand; the methionine responsible for interacting with the substrate in NAA50 (third position in β4 motif on Fig 4) is substituted by a glutamine in clusters 49 and 128 and by a glutamate in clusters 135 and 136, while sequences in cluster 212 retain the conserved methionine. Based on the presented differences we predict that clusters 49, 120, 128, 136, and 212 have substrate specificities different from that of NAA50. Sequences from clusters 49 and 120 were selected for experimental verification (See section 4.2 below).

Sequences in clusters 83 and 209 around cluster 24 (NAA60) (**Fig 2**) could display substrate specificities distinct from that of NAA60. The main difference is in the α1-α2 loop (**Fig 6A**); a conserved acidic residue in NAA60 is substituted by a conserved positive residue four residues downstream of the mutation-sensitive phenylalanine. Given the importance of the α1-α2 loop [49–51, 54] this is likely to result in a change in ligand specificity for clusters 83 and 209. Cluster 83 contains only plant sequences while cluster 24 contains only animal sequences. The differences in sequence motifs we observe could indicate differences in specificities between plant and animal NAA60.

In group 3, the component around cluster 20 (NAA40) also includes clusters 13, 47, 104 and 222 (**Fig 2**) which share NatD motifs to some extent. Crucial differences exist though.

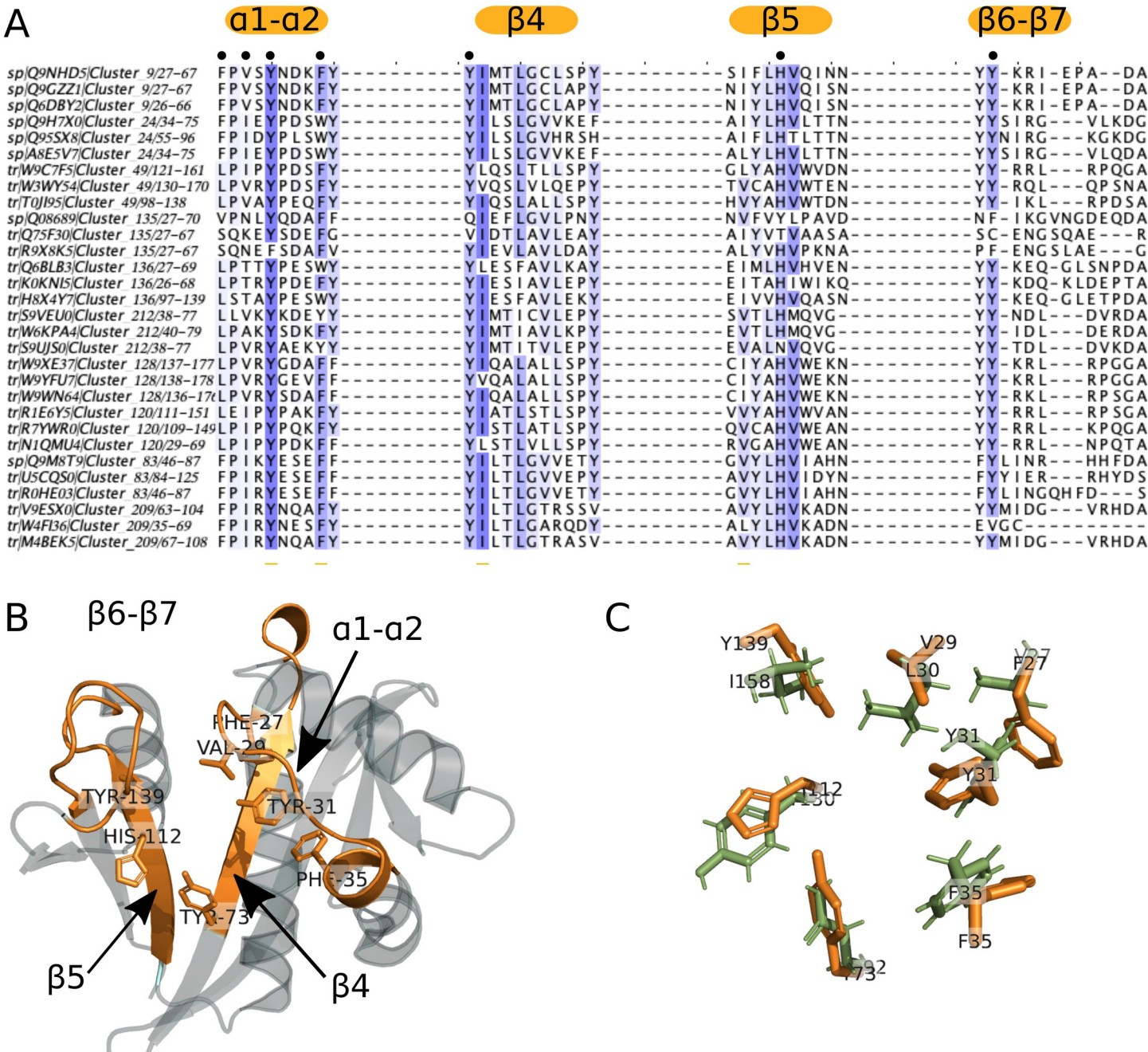

**Fig 6. Variations of sequence motifs in key positions on the GNAT fold suggest novel NATs with different ligand specificities.** When we compare the sequence motifs of NAA50 (cluster 9) and NAA60 (cluster 24) to the corresponding motifs of their surrounding clusters, we notice a number of small but meaningful differences (**A**). These differences occur on key positions of the GNAT fold and are illustrated here on the X-ray structure of NAA50 (PDB 3TFY) (**B**) The sequence differences located on the α1-α2 loop, β4 and β5 strands and β6-β7 loop residues are likely to result in altered specificity. The structure superimposition between human NAA50 from cluster 9 (orange, PDB 3TFY) and yeast NAA50 from cluster 135 (green, PDB 4XNH) highlights the small differences between residues involved in substrate binding in these two proteins with reportedly different specificities [88] (**C**).

Sequences in clusters 13, 104 and 222 lack an aspartic acid conserved in cluster 20 and located on the β3 strand of NAA40. Clusters 13, 47 and 222 lack a tryptophan conserved in the α1-α2 loop of NAA4 (S6C Text). Since both residues are crucial for substrate binding, the four

clusters around NAA40 might have different substrate similarities. In particular D127 in β3, together with Y136 and Y138 in β4, and E129 located between β3 and β4, is involved in interactions with the first 4 residues of the NatD substrate (H4 and H2A histones) and their mutation greatly reduces the catalysis rate [55]. Interestingly cluster 20 contains not only the well characterized animal NAA40 but also sequences from the plant phylum Chlorophyta. Other plant sequences found in this group form cluster 104 and belong to the phylum Streptophyta. Fungal sequences form cluster 47 and 222 and do not mix with plant and animal sequences. The observed differences between cluster 20 and 104 indicate that fungal, plant and animal NAA40 might have different specificities.

Even though there are four clusters around NAA80 (cluster 63, group 4) (**Fig 2**), we did not find any variations in their key sequence motifs (S6D Text). The clustering in this case was likely based on taxonomical differences.

**Experimental verification of clusters 49 and 120.**   To evaluate the accuracy of our predictions, we recombinantly expressed two candidates from the clusters 49 and 120, purified them and tested their ability to acetylate N-termini from a selection of 24 amino acids-long synthetic peptides (**Fig 7**). One of the candidate enzymes was N1Q410 from the fungus *Dothistroma septosporum*. After expression and subsequent purification of N1Q410 (S7A Text), we tested its ability to acetylate N-termini of different sequences in a DTNB-based spectrophotometric assay (**Fig 7A**). The first seven peptides represent typical substrates for the seven known NATs in higher eukaryotes (NatA, SESS; NatB, MDEL; NatC, MLPG; NatD, SGRG; NatE, MLGP; NatF, MLGP; NatH, DDDI) [7, 42]. The subsequent six peptides have been selected dependent on the initial results for both proteins, resembling amino acid combinations that are potential substrates. Although the overall activity of N1Q410 was relatively low, there was a clear preference for methionine starting peptides, especially MDEL (21.09 ± 4.03 μM) and MEEE (15.10 ± 0.25 μM) (**Fig 7A**). The putative NAT A0A194XTA9 from the fungus *Phialocephala scopiformis* (S7A Text) showed a higher activity in general as well as a broader substrate specificity (**Fig 7B**). Similar to N1Q410 only peptides starting with a methionine were Nt-acetylated by A0A194XTA9, with the peptides MAPL (50.92 ± 1.89 μM), MFGP (48.94 ± 2.50 μM) and MVEL (148.74 ± 2.25 μM) showing the highest activities.

Group 2 also harbors clusters 9 and 24 containing known NATs NAA50 and NAA60, respectively. Thus, we would expect that proteins from other Group 2 clusters would express NAT-activity and further that these display a substrate specificity similar to what is observed for NAA50 and NAA60. NAA50 and NAA60 have overlapping substrate specificities *in vitro*,

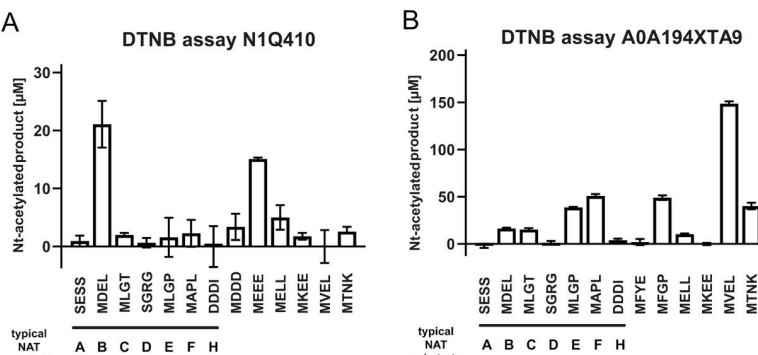

**Fig 7. Purification and DTNB-based activity assays of the putative NATs N1Q410 and A0A194XTA9.** Putative NAT activities were tested by DTNB-based assays. 3μM of purified N1Q410 (**A**) and A0A194XTA9 (**B**) were incubated with a selection of 24 amino acids-long synthetic peptides (300 μM), and Ac-CoA (300 μM) for 1 hour at 37˚C. The formation of Nt-acetylated product was spectrophotometrically determined. Shown is the mean ± SD (n = 3).

but *in vivo* substrates are not likely to overlap since NAA50 is nuclear/cytosolic and partly anchored to the ribosome via NAA15-NAA10 [89, 90] while NAA60 acetylates transmembrane proteins via anchoring to the cytosolic side of the Golgi-membrane and other cellular membranes [33, 91]. Both enzymes may acetylate a variety of Met-starting N-termini, in particular Met-Leu, Met-Ala, Met-Val, Met-Lys, Met-Met [38, 92].

Both N1Q410 and A0A194XTA9 display clear N-terminal acetyltransferase activity confirming that these are true NATs (**Fig 7**). Furthermore, both enzymes prefer Met-starting N-termini among the peptides tested. A0A194XTA9 has a clear preference for the NatE/NatF (NAA50/NAA60) type of substrates strongly suggesting that this NAT is either a NAA50 or NAA60 type of enzyme in *P. scopiformis*. For N1Q410, we observe a preference for N-terminal peptides where Met is followed by an acidic residue at the second position, very similar to NAA20/NatB activity [39] despite the fact that it harbours sequence motifs highly similar to those of Group 2 NATs. This is an example of how sensitive N-acetyltransferase ligand specificity can be to subtle sequence changes. In this case they mainly consist in: (1) two substitutions in the α1-α2 loop where the highly conserved F and V in the NAA50 motif are replaced by an L and an I, respectively, (2) NAA50 has only one proline in the α1-α2 loop while cluster 120 has 2 and (3) the highly conserved M in the β4 strand of NAA50 is at a different position in cluster 120 sequences. Thus, N1Q410 might be a NAA20 type enzyme which is clustered among NAA50/NAA60 type enzymes in Group 2, or there might be other factors skewing the substrate preference *in vitro*.

## Conclusion

Using pairwise sequence comparisons and phylogenetic analyses we have mapped the sequence space of the eukaryotic N-terminal acetyltransferases superfamily and the evolutionary relationship between its members.

### NATs can be grouped into 5 groups in the highly diverse acetyltransferase superfamily

Using information from the network topology together with the identification of sequence motifs for each of the known NATs, we could classify NATs into 5 different groups: Groups 1 to 5. Even though NAA10 and NAA20 do not cluster together with NAA30 in the SSN, we chose to place them in the same group because of the similarity between their sequence motifs. Group 1 thus consists of two subgroups: Group 1a (NAA10, NAA20) and Group 1b (NAA30). Groups 1a and 1b share a lot of similarities with Group 2 (NAA50 and NAA60) such as their sequence motifs. Common ancestry of groups 1 and 2 is supported by the conserved sequence motifs which interestingly do not all necessarily retain a significant functional role in each of the NAT groups. A less stringent SSN also shows closer clustering of the Group 1 and Group 2 NATs. Our data also highlights the low sequence similarity between NAA40 and NAA80 on one hand, and the other NATs on the other hand.

A recent study that demonstrates N-terminal acetylation activity of eight enzymes in *A. thaliana* plastids [61] indicate the possible existence of more than 5 groups of NATs. We refrain from assigning more NAT groups based on this most recent discovery since the ability to acetylate N-termini does not necessarily mean N-terminal acetyltransferase function.

### Different evolutionary paths

Each of the NAT groups have clear characteristics that distinguish them unequivocally from one another. Our results indicate that N-terminal acetyltransferases evolved more than once on the GNAT fold. The phylogenetic tree, which informs on the position of different NATs in

the acetyltransferase superfamily and provides a useful perspective on the differences between ligand specificities, confirms this. The relationships between enzymes revealed by the SSNs and the structural comparison are also in agreement with the phylogenetic tree. Interestingly NAT groups are found on the same branches as acetyltransferases known to have other functions. N-terminal acetyltransferases are not one homogenous, uniform, family of enzymes and the GNAT fold has evolved different specificities more than once.

## Consequences for function and functional annotation of acetyltransferases

Because of this we cannot exclude that N-terminal acetyltransferases can acetylate other substrates than N-terminal amines. NAA10 and NAA60 are suspected to be able to acetylate lysine side chains in addition to protein N-termini [93, 94], even if this has been debated [95]. A related consequence is that other acetyltransferases might be able to acetylate N-terminal amino acids. One of the most recent findings is that glucosamine 6-phosphate acetyltransferases can acetylate protein N-termini [60]. Moreover, our results indicate that serotonin acetyltransferases could have the ability to acetylate protein N-termini and have a biological role as N-terminal acetyltransferases, as well, which has been indicated in a recent finding that *A. thaliana* serotonin acetyltransferase has weak N-terminal acetylation ability [61]. This is relevant in the quest and characterization of yet-to-be discovered enzymes catalyzing N-terminal acetylation of particular groups of protein N-termini (for instance those resulting from post-translational protease action) or specific proteins (analog to NAA80 specifically acetylating actins). Indeed, the currently known NATs are not yet defined as responsible for all cellular N-terminal acetylation events though the major classes of co-translational acetylation have been accounted for using *S. cerevisiae* genetics and proteomics [11, 39, 47]. In the human proteome we could not find uncharacterized sequences qualifying as NATs as per the characteristics we define in this study. It is therefore important to thoroughly inspect all close relatives to known NATs for the discovery of new enzymes.

The fact that there is not one single catalytic site and mechanism for acetylation even for the closest of NATs creates another conundrum. NAA10, for example, has a conserved glutamate in α1-α2 loop which is involved in catalysis, but in the case of NAA20, NAA10's closest relative, the same conserved glutamate has no role in catalysis [51, 54]. This case became even more puzzling when the study of NAA20 revealed no obvious catalytic residue. Furthermore, NAA10 acetylates different substrate N-termini when in a monomeric form as compared to when it is complexed with its auxiliary subunit NAA15 [54, 92]. It can look as if as long as a substrate can bind properly to the GNAT, the chances are high it can be acetylated. It follows that the impossibility to strictly define what makes N-terminal acetyltransferases acetylate N-termini and no other substrates greatly limits our ability to predict NAT function from sequence. We are left to only comparing key sequence motifs in order to detect similarities and predict NAT function. Yet, subtle sequence changes might also affect substrate specificity. Despite those difficulties we were able, using this approach, to predict two new NATs and confirm their function by acetylation assays *in vitro*.

## About the use of the classification for functional annotation of uncharacterized sequences

Representatives from the clusters 49 and 120 from Group 2, N1Q410 from the fungus *D. septosporum* and A0A194XTA9 from the fungus *P. scopiformis* were expressed, purified and subjected to *in vitro* NAT assays. A0A194XTA9 has a clear preference for the same type of substrates as NAA50/NAA60 but the substrate specificity of N1Q410 resembles that of

NAA20, highlighting how sensitive the ligand specificity of NAA is to subtle sequence substitutions.

The superfamily has highly diverged in primary structure, but secondary and tertiary structures remain largely intact. The GNAT is the scaffold on which numerous types of molecules can get acetylated and it evolves different specificities by changes in sequence that do not affect the overall structure. Our work shows that it is possible, within the limits discussed in "Consequences for function and functional annotation of acetyltransferases", to predict ligand specificity similarity or differences between GNAT-containing sequences if they are closely related and by comparing the key sequence motifs that we report here. Predicting the substrate specificity of an uncharacterized GNAT sequence which doesn't have close relatives with known function is practically impossible *in silico*. *In vitro* assays are necessary to map function and specificity of uncharacterized parts of the acetyltransferase superfamily. It is important to note that large portions of the phylogenetic tree have exclusively uncharacterized sequences and it is impossible to say anything about their substrate specificity. There are no human proteins in the uncharacterized parts of the tree. While this work is restricted to eukaryotic GNAT-containing sequences and encompasses the majority of eukaryotic acetyltransferases it is important to mention that some non-GNAT acetyltransferases like FrBf [96] were discovered as recently as in 2011. Members of the MYST family [97] are also relevant non-GNAT acetyltransferases. New potential acetyltransferases could be found among those enzymes. Moreover recent studies have shown that most N-terminal acetyltransferases evolved before eukaryotic cells [52] so it might be that looking at bacterial and archaeal proteomes would provide valuable information.

In summary our work provides the first classification and phylogenetic analysis of the eukaryotic GNAT acetyltransferases superfamily. It reveals that NATs evolved more than once on the GNAT fold and that they do not form a homogenous family. We provide sequence motif signatures of known NATs that, together with this classification form a solid basis for functional annotation and discovery of new NATs.

## Material and methods

### Sequence similarity networks (SSN)

**Collection of sequence dataset.** All members of GCN5-related acetyltransferase superfamily contain the GNAT fold. As there is no finer classification to aid dataset creation, we retrieved all UniProt sequences that match the GNAT fold signature as defined by PROSITE [64, 98]. According to PROSITE there are four types of GNAT fold–GNAT (code: PS51186), GNAT_ATAT (code: PS51730), GNAT_NAGS (code: PS51731) and GNAT_YJDJ (code: PS51729). These PROSITE signatures match sequences from all domains of life (around 900 000 sequences in UniProt). We restricted our dataset to only eukaryotic entries (more than 50000 sequences) in agreement with the focus of this work. We kept all SwissProt (manually curated) sequences and *Homo sapiens* TrEMBL (not reviewed) sequences in the dataset. The remaining TrEMBL sequences were filtered to reduce the size of the dataset. Filtering of TrEMBL sequences was performed using h-cd-hit [70, 99] in three steps–a first run performed at 90% identity, a second at 80% and a third at 70% identity. The threshold was set to be 70% sequence identity as this usually indicates shared function [100–102]. We created two datasets using this strategy: the full-sequence dataset (S1 Dataset) and the GNAT-domain dataset (S2 Dataset). We used the pfamscan tool from Pfam [103] together with HMMER3.2.1 [104] to locate the GNAT fold boundaries in the full-sequence dataset in order to generate the GNAT-domain dataset.

**Generating the SSNs.** The final, filtered, dataset (14396 sequences) was used to generate the SSN using EFI-EST [71] with the following parameters: E-value of $10^{-15}$ and alignment

score of 30 (S7 File). The chosen values ensured that sequences clustering together were closely related (Cf S1A Text) with a minimal sequence identity equal to 40% yielding isofunctional clusters. The shortest sequence kept in the network was 34 amino acids long. It is not known yet what the minimal functional part of the GNAT fold is. The resulting network was analyzed using Cytoscape [73]. To visualize the network in Cytoscape we used γfiles organic algorithm by γWorks (https://www.yworks.com/). In addition to the network made from E-value thresholds equal to $10^{-15}$ and alignment score equal to 30 we created several other networks, mainly for the purpose of finding the best dataset for phylogenetic analyses (see Phylogeny section below for more details). Parameters for these networks were: for E-value of $10^{-5}$, alignment scores of 15, 20, 25, 30, 35 or 40; for E-value of $10^{-10}$, alignment scores of 15, 20, 25, 30, 35 or 40; for E-value of $10^{-15}$, alignment score 16, 25, 30, 35 or 40; for E-value equal to $10^{-20}$, alignment score equal to 20, 30, 35 or 40.

**Identification of isofunctional clusters and their neighbours.**   In the resulting SSN (E-value $10^{-15}$ and alignment score 30) there were no clear boundaries between different clusters. In order to identify separate clusters, we applied the clusterONE algorithm [72] which is designed to recognize dense and overlapping regions in a graph. The search for dense regions in a network (clusters) was performed with the following parameters: minimum size of 10 sequences for a cluster to be considered, minimum density: auto, edge weights: percentage identity, and the remaining settings were taken at their default values. Next, we identified known NATs, and other non-NAT acetyltransferases, in their corresponding clusters (using annotation details added to the network) and we let these clusters be defined by experimentally confirmed enzymes (based on the assumption of cluster isofunctionality). Given the high percentage identity inside the identified clusters, we assumed cluster isofunctionality (i.e. similar ligand specificity) and transferred annotation from experimentally confirmed proteins to unknown ones within the same cluster. We also created a simplified network using the clusterONE results as input. We represented each cluster defined by ClusterONE as a single node. Nodes in the simplified network are connected by an edge if at least one edge exists between nodes of two given clusters in the original network. After adding all nodes and edges to the simplified network, we applied γFiles (https://www.yworks.com) orthogonal algorithm to get the final view.

**Network analyses.**   The topology of the simplified network was analyzed using Network Analyzer through Cytoscape. Mainly, we used node degree and betweenness centrality, where node degree tells how many neighbors a node has and betweenness centrality describes how important is a given node for interactions between different parts of a network. Network analyzer calculates betweenness centrality using the algorithm by Brandes [105]. Betweenness centrality was calculated on the representative, "pivot" network which is not weighted and not directed.

**Motif discovery.**   We used the MEME tool [74] to find characteristic sequence motifs within clusters. Each motif search was performed on all sequences of a given cluster. Enriched motifs were discovered relative to a random model based on frequencies of letters in the supplied set of sequences. As we work with protein sequences, zero to one occurrence of each motif per sequence was expected and searched for. A maximum of 25 unique motifs were searched for per sequence set, with 5 to 10 amino acid width. Only motifs with e-value below 1 were taken onto account. All motifs were visualized using SequenceLogoVis [106].

## Prediction of NATs among uncharacterized sequences

The prediction of NATs among uncharacterized sequences in the SSN started by the selection of the 29 clusters (cluster numbers: 227, 3, 121, 42, 62, 36, 82, 40, 114, 25, 223, 168, 135, 136,

212, 128, 49, 120, 83, 209, 106, 232, 104, 226, 104, 13, 222, 46, 47) neighboring the clusters containing known NATs, namely clusters 10 (NAA10), 8 (NAA20), 2 (NAA30), 20 (NAA40), 9 (NAA50) 24 (NAA60), 97 (NAA70) and 63 (NAA80). We searched for occurrences of key sequence motifs of known NATs (shown in Fig 4 of the Results section) in all sequences of the 29 selected clusters using MAST [107]. When we found in a cluster sequence motif similar to that of a cluster of a known NAT, we generated a MSA using three random sequences from the identified cluster and three sequences from the cluster of known NATs.

## Structural comparison

We used the DALI server to perform an all-by-all structural comparison of 38 unique structures of Eukaryotic acetyltransferase catalytic subunits. The comparison was done between selected intramolecular distances and no sequence information was used. The subunits are ranked based on the calculated pairwise similarities (Dali Z-scores) and a dendrogram that depicts the ranking was drawn [80, 81, 108]. The resulting dendrogram is the result of hierarchical clustering of structures based on the matrix of Dali Z-scores. The list of PDB codes used for structural comparison is available in S6 File.

## Phylogeny

**Choice of sequence dataset for phylogeny.** Since there are no clear boundaries between different acetyltransferases, due to lack of detailed classification, we based our phylogeny analysis on our SSNs. We used the more stringent SSN (E-value = $10^{-15}$, alignment score = 30) and selected three representative sequences for each cluster. In order to create the dataset for phylogenetic analyses, we created several networks that allowed for more connections between nodes (and clusters) (see Table A in S4 Text) and looked for the SSN with the largest single connected component (the largest group of clusters) exhibiting smallworld properties [109]. We calculated smallworldness for each of the largest connected components using NetworkX Python library [110].

**Sequence alignment for phylogeny.** We selected three sequences per cluster to generate the multiple sequence alignment (S2 File). If a cluster contained sequences from SwissProt, those sequences were used in the alignment. Otherwise, TrEMBL sequences were randomly selected as cluster representatives. As sequence divergence within the acetyltransferase superfamily is extremely high, we used only the highly conserved A and B motifs of the GNAT fold. The alignment was generated using Clustal Omega [111] and the full alignment was constructed step by step. Sequences from closely related clusters were aligned first and different alignments were then merged using MAFFT [112]. Merging two alignments using MAFFT was always performed using "anchor" sequences and ensuring that both alignments had one set of five sequences (i.e. one cluster) in common ("anchor" sequences). That also ensured that corresponding secondary structure elements was kept intact after merging. Alignments generated for merging were manually edited, using acetyltransferases with known structures used as reference to increase the alignment precision.

**Model of evolution.** To select the right amino acid replacement model, which describes the probabilities of amino acid change in the sequence, we used ProtTest3 [113]. As input, we used the previously generated multiple sequence alignment. Tested substitution model matrices were JTT [114], LG [115], DCMut [116], Dayhoff [117], WAG [118] and VT [119]. All rate variations were included in the calculation (allowing proportion of invariable sites or +I [120], discrete gamma model or +G [121] (with 4 rate categories) and a combination of invariable sites and discrete gamma model or +I+G [122]. Empirical amino acid frequencies were used. We calculated a maximum likelihood tree to be used as starting topology.

**Construction and evaluation of the phylogenetic tree.**   Finally, a maximum likelihood tree was calculated using RAxML [123] based on the generated alignment (S1 File). We used LG+G+F model of evolution since it provided the best fit according to prottest3 [113] calculation (with AIC, AICc and BIC models selection strategies). Ten searches for the best tree were conducted. Once the best tree was calculated, its robustness was assessed using bootstrap. As stop criterion we used a frequency-based criterion, by calculating the Pearson's correlation coefficient [124]. After bootstrapping was complete, we used transfer bootstrap expectation (TBE) [125] which has been shown to be more informative than Felsenstein's bootstrap method for larger trees built with less similar sequences.

## Experimental

A detailed description of the material and methods is provided in S7 Text. In brief, the genes *N1Q410* and *A0A194XTA9* were cloned into pETM11 vectors. The encoded proteins were recombinantly expressed in *E. coli* BL21 Star (DE3) cells and purified using affinity and size exclusion chromatography. The purity of the proteins was determined by SDS-PAGE and protein concentrations were determined spectrometrically. The enzyme activity was determined via DTNB assay as described in [126].

## Supporting information

**S1 Text. General information about network construction and network topology.**
(PDF)

**S2 Text. Five groups of known N-terminal acetyltransferases.**
(PDF)

**S3 Text. Multiple sequence alignments in the N-terminal acetyltransferase family.**
(PDF)

**S4 Text. Small-world SSN.**
(PDF)

**S5 Text. Phylogenetic analyses.**
(PDF)

**S6 Text. Network analyses: predictions of new NATs.**
(PDF)

**S7 Text. Experimental verification of clusters 49 and 120.**
(PDF)

**S1 Fig. Structure similarity networks constructed based on similarity matrix obtained from DALI.** Acetyltransferases are structurally highly similar, which is reflected in the similarity networks and how they change as a function of the threshold Z-score. Networks constructed with Z-score > 2 and Z-score > 10 are random. First separation between nodes into different clusters is at Z-score > 15.
(TIF)

**S1 Dataset. Sequences used to calculate the full-length sequence SSN, given in the FASTA format.**
(TXT)

**S2 Dataset. Sequences used to calculate SSNs for the GNAT domain portion of sequences.** All sequences are provided in the FASTA format.
(TXT)

**S3 Dataset. Cluster 97 sequences.** Sequences in FASTA format found in cluster 97 of our full-sequence SSN. These sequences belong to the NAA70 plastid N-terminal acetyltransferase and were used as the dataset for calculating Group 5 sequence motifs.
(TXT)

**S1 File. Phylogenetic tree.** Phylogenetic tree of the GNAT acetyltransferase superfamily calculated using RAxML in Newick format. Leaves of the tree are labeled with protein accession number and a corresponding cluster number. Inner nodes of the tree are labeled with calculated support values for each node.
(TXT)

**S2 File. Multiple sequence alignment used for calculating the phylogenetic tree.**
(TXT)

**S3 File. Sequence motifs for Group 5 of NATs calculated using MEME tool from MEME Suite.** The file contains 1) motif P-values; 2) block diagrams showing the position of the motifs on the relevant sequences; 3) PSSM; 4) position-specific probability matrix; 5) regular expression for the given motif.
(TXT)

**S4 File. Positions of Group 5 sequence motifs on the representative sequence calculated using MAST from the MEME Suite.**
(TXT)

**S5 File. List of members of the SSN and their corresponding cluster numbers.** This is the table of all proteins from our SSN. The table contains accession numbers, Uniprot annotation status (SwissProt/TrEMBL), description and a corresponding cluster number for each of the proteins.
(XLS)

**S6 File. List of PDB structures.** List of all PDB structures used for structural comparison with DALI. The list contains PDB codes, names of chains and protein names that were used for comparison.
(XLSX)

**S7 File. ZIP file of SSN in xgmml format.** SSN calculated based on the full-length sequence acetyltransferase dataset. Available also at http://doi.org/10.5281/zenodo.4288938
(ZIP)

## Acknowledgments

We thank Angèle Abboud for her insightful comments on our manuscript.

## Author Contributions

**Conceptualization:** Thomas Arnesen, Nathalie Reuter.

**Data curation:** Bojan Krtenic.

**Formal analysis:** Bojan Krtenic, Adrian Drazic, Thomas Arnesen, Nathalie Reuter.

**Funding acquisition:** Thomas Arnesen, Nathalie Reuter.

**Investigation:** Bojan Krtenic, Adrian Drazic.

**Methodology:** Bojan Krtenic, Adrian Drazic.

**Project administration:** Thomas Arnesen, Nathalie Reuter.

**Resources:** Thomas Arnesen, Nathalie Reuter.

**Software:** Bojan Krtenic.

**Supervision:** Thomas Arnesen, Nathalie Reuter.

**Validation:** Bojan Krtenic, Adrian Drazic, Thomas Arnesen, Nathalie Reuter.

**Visualization:** Bojan Krtenic, Nathalie Reuter.

**Writing – original draft:** Bojan Krtenic, Nathalie Reuter.

**Writing – review & editing:** Bojan Krtenic, Adrian Drazic, Thomas Arnesen, Nathalie Reuter.

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
