## [Decision Letter · Decision Letter 0]

17 Jun 2020

Dear Prof. Reuter,

Thank you very much for submitting your manuscript "Classification and phylogeny for the annotation of novel eukaryotic GNAT acetyltransferases" for consideration at PLOS Computational Biology.

As with all papers reviewed by the journal, your manuscript was reviewed by members of the editorial board and by several independent reviewers. In light of the reviews (below this email), we would like to invite the resubmission of a significantly-revised version that takes into account the reviewers' comments.

We cannot make any decision about publication until we have seen the revised manuscript and your response to the reviewers' comments -- in particular those by reviewer #1 who was the most critical. Your revised manuscript is also likely to be sent to reviewers for further evaluation.

Sincerely,

Christos A. Ouzounis

Associate Editor

PLOS Computational Biology

Arne Elofsson

Deputy Editor

PLOS Computational Biology

Reviewer's Responses to Questions

**Comments to the Authors:**

Reviewer #1: * What are the main claims of the paper and how significant are they for the discipline?

Krtenic et al’s manuscript describes how they have classified the eukaryotic protein N-terminal

acetyltransferases (NATs) into five groups using sequence similarity networks and characterized

their sequences in terms of motif presence and phylogeny. I’m sure their classification is useful in

the field, but a wider scope, primarily by including evolutionarily related sequences from bacteria

and archaea would have made their contribution much more significant.

* Are these claims novel? If not, which published articles weaken the claims of

originality of this one?

Yes.

* Are the claims properly placed in the context of the previous literature?

Have the authors treated the literature fairly?

Yes.

* Do the data and analyses fully support the claims? If not, what other evidence is required?

Not fully. The major shortcoming with the manuscript is the limitation to eukaryotic sequences.

Besides reducing the generality of the findings, all genes in eukaryotes do not have the same

evolutionary history -- some were likely vertically transferred from the last eukaryotic ancestor

(and is hence likely shared with some modern Archaea), others have been introduced horizontally via

the mitochondrion or chloroplast or other HGT events. It is possible that including bacterial and

archaeal sequences would reveal further events like this besides the group 5 that the authors note

is of bacterial origin. Without this, many of the claims in the manuscript are not very conclusive.

(I note that the authors make the same suggestion, to include bacterial and archaeal sequences, in

lines 602-4.)

Another shortcoming is the very limited information available in the alignment (75 amino acids,

including several gappy positions) on which the authors build their phylogeny. There is possibly not

much the authors can do about this, but if this is actually the case, perhaps a trustworthy

phylogeny is not possible or the authors could try to show how robust their phylogeny is by adding

some more positions to the alignment even though they might not be conclusively aligned (here I

would also recommend using a tool to select trustworthy positions, e.g. BMGE (Criscuolo and Gribaldo

2010)) plus looking at possible conflicts in the data by e.g. phylogenetic networks. If it proves

impossible to show that the tree is robust and hence is not trustworthy, it might be better to base

the classification on sequence similarity alone. Moreover, despite the tree not being rooted, the

authors make a number of claims for the evolution of different groups of sequences which, strictly

speaking, are not possible to do based on an unrooted tree.

Furthermore, in the phylogeny, group 1 is clearly not monophyletic. Although the authors make no

claim to discover evolutionarily related groups, I think that it would be good to, after noting that

the group is not monophyletic, revisit the classification to see if there is evidence in the SSN and

motif search to split the group into two. The description of this might be easier to follow in a

more reduced result section where all evidence for potential groups is first presented, then

followed by a proposal (but see above for more comments regarding the disposition of the material)

which in turn is followed by a discussion section.

* Would additional work improve the paper? How much better would the paper be if this work were

performed and how difficult would it be to do this work?

Potentially yes. Extend data with bacterial and archaeal sequences. This is of course a major

undertaking, but would make results interesting for a larger group of readers as well as potentially

reveal interesting evolutionary histories of some of the proteins. If the above mentioned problems

with the phylogeny and alignment can be handled in a convincing way, adding sequences would

potentially also allow the eukaryotic part of the tree to be rooted and stronger claims could be

supported.

* PLOS Computational Biology encourages authors to publish detailed protocols and algorithms as

supporting information online. Do any particular methods used in the manuscript warrant such

treatment? If a protocol is already provided, for example for a randomized controlled trial, are

there any important deviations from it? If so, have the authors explained adequately why the

deviations occurred?

Not applicable.

* Are original data deposited in appropriate repositories and accession/version numbers provided for

genes, proteins, mutants, diseases, etc.?

No original data is presented and the authors have done a good job providing results in machine

readable format as supplementary material.

* Does the study conform to any relevant guidelines such as CONSORT, MIAME, QUORUM, STROBE, and the

Fort Lauderdale agreement?

Not applicable.

* Are details of the methodology sufficient to allow the experiments to be reproduced?

Yes.

* Is any software created by the authors freely available?

There is no author written software mentioned in the manuscript.

* Is the manuscript well organized and written clearly enough to be accessible to non-specialists?

Partly. The results section is written like a combined results and discussion section, then follows

a second discussion section. I can see that this kind of manuscript is quite well suited for a

combined results and discussion section, although the presentation of the phylogeny, which throws

part of the classification in question (group 1 is not monophyletic), calls for a revisit of the

grouping which might be more amenable to a shorter result section followed by a discussion section.

* Have any parts of the paper been published elsewhere? Are there any copyright issues associated

with this that conflict with the PLOS license? If this is the case, please alert the journal

office by email.

No.

* Does the paper use standardized scientific nomenclature and abbreviations? If not, are these

explained at the first usage?

Yes

* Comments to the authors:

As I have rather wide reaching suggestions for the content as well as presentation of the manuscript

above, below follows a non-exhaustive list of specific comments.

L29, 51 & 125: “homology” should be replaced with “sequence similarity”.

L35: “fungi” should be “fungal”.

L70 & 168: “higher” and “lower” are unfortunate words in evolutionary discussions. What do you

actually mean: multicellular eukaryotes or what?

L106: “The majority” not “Majority”.

Figure 2: The colours of the groups in the legend to the right are not the ones used in the figure.

L206: “which extends over and covering the binding site” should be “extends over and covers”

L210: “has evolved to catalyzes N-terminal” should be “to catalyze”

L225: Should “length” be “height”?

L304: The manuscript did not specify if KAT14 has the Tyrosine involved in substrate binding

(located in β6-β7 loop and present in group 1 and group2 NATs). It is then not clear if KAT14 might

accommodate the same type of substrate or just a large dimension substrate (thanks to the presence

of the β6-β7 loop).

L517: Since the phylogenetic tree is based on a MSA that excluded the ligand specificity motifs

(such as α1-α2 or β6-β7 loop), except for groups 3 and 4 (β4 and β5), it is not accurate to affirm

that : “The phylogenetic tree (...) provides a useful perspective on the evolution of ligand

specificities”.

L581: “where the highly conserved F and V in the of NAA50 motif are replaced” : missing word? Or

just extra “of”?

L633: “minimal sequence identity equal to of 40% on average” should be “equal to 40%”

L659: It is useful to mention if the network used here is weighted or unweighted (and what defines

the edge weight), as the calculation of betweenness centrality is different.

Reviewer #2: This paper performed a superfamily wide bioinformatic characterization of the GCN5-related N-acetyltransferase (GNAT) superfamily. The bioinformatic pipeline that they used in this study sounds and provide a useful picture of the GNAT superfamily. I believe that the manuscript is suitable for publishing PLoS computational biology. I have only several suggestions to revise the manuscript to reach to wider communities.

It is nice if the authors labeled other nodes with known enzyme function (in Figure 2, Figure 5 and related supplementary figures). I understood the focus of the work is studying NAT groups within the superfamily. However, mapping other functions would make this paper far more impactful and useful as it reaches to researchers that are working on non-NAT acetyltransferase in the superfamily. There are some in Figure 5, but it seems that there are more (as far as I can see in Figure 4). It would be nice if the authors showed all characterised clusters (regardless NAT or other substrates), and unexplored clusters.

It is unclear the relationships between Figure 2 and 5. The should label cluster names as much as possible in Figure 5 to make it clear (in particular black lines).

Similarly, it is informative if the authors label each PDB structure for associated cluster in the SSNs (Figure 2).

Structural comparison - it is unclear how the authors compared the structure and what is the meaning of phylogeny (Figure 4). There is no description is available in Methods (just mentioned that used Dali Server). The authors should provide more detailed description in the text and Method, and explain the implication of the structural phylogeny, what is based on RMSD? what is the meanings of nodes and branches?

Again, I understood that the focus of the work is NAT, it would be nice to have a small discussion about how much clusters are uncharacterised and what would be substrates for those clusters.

Reviewer #3: The manuscript entitled “Classification and phylogeny for the annotation of novel eukaryotic GNAT acetyltransferases”describes a state-of-the-art computational approach for the functional annotation of the protein acetyltransferases that belong to the superfamily of GNAT acetyltransferases.

Their approach classifies the huge number of protein acetyltransferases into subgroups based upon specific structural features. This classification will be useful in future studies to predict and dissect the manifold acetyltransferases present in eukaryotes. The authors also provide a proof-of-concept for the predictive power of their classification system.

However, a couple of minor concerns should be addressed prior to publishing of the study:

Introduction

- L73: … has been shown that N-terminal acetylation affects ….

In this context, the very clear connection between N-terminal acetylation and diverse abiotic and biotic stress responses in plants should be mentioned

(Linster et al., 2015; Xu et al., 2015; Armbruster et al., 2020; Huber et al., 2020; Neubauer and Innes, 2020) reviewed in (Linster and Wirtz, 2018)

- L98 -: The NATs can be more or less promiscuous when it comes to substrate specificity (7).

This statement is not wrong but misleading to the non-expert reader. Please specify, “With respect to substrate specificity, some Nats are more promiscuous than others, e.g. NatB from fungi, plants and human share a narrow substrate specificity (MD, ME, MQ, MN) while other Nats possess more relaxed e.g. plant NatG or human NatF (Refs).

Here, the conservation of substrate specificities for distinct Nats should be introduced! Otherwise, the classification with respect to the catalytic subunits (Fig 2) makes not much sense.

Results

- The authors nicely compare features from animal and fungal Nats, but neglect the importance of the N–acetyltransferase machinery in plants. With the exception of NatG, the well characterized co-translationally acting Nats of the reference plants Arabidopsis thaliana are not mentioned throughout the result section.

- L185: 26-NAA50 should be 24-NAA50 according to the Figure. Does this cluster also contain the inactive yeast NAA50?

- L 196: … “we defined five different groups of NATs (Fig 2).” This is misleading since Figure 2 shows only 4 groups.

- L 199: Group 1 contains NAA10, NAA20 and NAA30. NAA10 and NAA20 are in the same connected component, while NAA30 is found in a single isolated cluster.” If NAA30 is part of the group 1, it should be shown in Fig2

- L 203 “Group 2 consists of NAA50 and NAA60” Does the group also consist of catalytically inactive NAA50 from yeast? Please discuss.

- - L 233 “This tyrosine is essential for function and is strictly conserved in all members of groups 1 and 2 (43–45,48,68).” Please include information about the origin of catalytic subunits (fungi, plantae, animalia)

- L 268: 4u9vA_NAA40-Q86UY6 is separated from group 1-4. This contradicts the main message of figure 4. Is there any explanation why it is so different from 4u9wA_NAA40_substrate? E.g., lower resolution of structure, or artificial crystallization conditions -> if so, please skip

- L 298 Please also discuss the yeast NAA50 in this respect.

References:

Armbruster, L., Linster, E., Boyer, J.-B., Brünje, A., Eirich, J., Stephan, I., Bienvenut, W.V., Weidenhausen, J., Meinnel, T., Hell, R., Sinning, I., Finkemeier, I., Giglione, C., and Wirtz, M. (2020). NAA50 is an enzymatically active Nα-acetyltransferase that is crucial for the development and regulation of stress responses. Plant Physiology, accepted.

Huber, M., Bienvenut, W.V., Linster, E., Stephan, I., Armbruster, L., Sticht, C., Layer, D., Lapouge, K., Meinnel, T., Sinning, I., Giglione, C., Hell, R., and Wirtz, M. (2020). NatB-Mediated N-Terminal Acetylation Affects Growth and Biotic Stress Responses. Plant Physiology 182, 792-806.

Linster, E., and Wirtz, M. (2018). N-terminal acetylation: an essential protein modification emerges as an important regulator of stress responses. J Exp Bot 69, 4555-4568.

Linster, E., Stephan, I., Bienvenut, W.V., Maple-Grodem, J., Myklebust, L.M., Huber, M., Reichelt, M., Sticht, C., Geir Moller, S., Meinnel, T., Arnesen, T., Giglione, C., Hell, R., and Wirtz, M. (2015). Downregulation of N-terminal acetylation triggers ABA-mediated drought responses in Arabidopsis. Nat Commun 6, 7640.

Neubauer, M., and Innes, R.W. (2020). Loss of the Acetyltransferase NAA50 Induces ER Stress and Immune Responses and Suppresses Growth. Plant Physiology, pp.00225.02020.

Xu, F., Huang, Y., Li, L., Gannon, P., Linster, E., Huber, M., Kapos, P., Bienvenut, W., Polevoda, B., Meinnel, T., Hell, R., Giglione, C., Zhang, Y., Wirtz, M., Chen, S., and Li, X. (2015). Two N-terminal acetyltransferases antagonistically regulate the stability of a nod-like receptor in Arabidopsis. Plant Cell 27, 1547-1562.

**Have all data underlying the figures and results presented in the manuscript been provided?**

Reviewer #1: Yes

Reviewer #2: Yes

Reviewer #3: Yes

PLOS authors have the option to publish the peer review history of their article (what does this mean?). If published, this will include your full peer review and any attached files.

Reviewer #1: No

Reviewer #2: No

Reviewer #3: No
---

## [Decision Letter · Decision Letter 1]

10 Aug 2020

Dear Prof. Reuter,

Thank you very much for submitting your manuscript "Classification and phylogeny for the annotation of novel eukaryotic GNAT acetyltransferases" for consideration at PLOS Computational Biology.

As with all papers reviewed by the journal, your manuscript was reviewed by members of the editorial board and by several independent reviewers. In light of the reviews (below this email), we would like to invite the resubmission of a significantly-revised version that takes into account the reviewers' comments.

We cannot make any decision about publication until we have seen the revised manuscript and your response to the reviewers' comments. Your revised manuscript is also likely to be sent to reviewers for further evaluation.

Sincerely,

Arne Elofsson

Deputy Editor

PLOS Computational Biology

Arne Elofsson

Deputy Editor

PLOS Computational Biology

Reviewer's Responses to Questions

**Comments to the Authors:**

Reviewer #1: Overall assessment

I confess I was confused when performing my first review of Krtenic et al.’s manuscript, and I still am, despite the efforts of the authors in this revised manuscript. Let me summarise how I interpret the data presented to hopefully clarify any further misunderstandings. 1) Eukaryotic NAT sequences were clustered based on similarity. Some of the identified clusters contain members of known function (named NAAnn). 2) Each cluster was searched for sequence motifs which served as the basis for grouping some clusters by identification of similarities in motifs. 3) To investigate if the sequence similarity clustering is reflected in structural similarity, a distance-based phylogenetic tree was built from structural similarity scores. 4) A maximum likelihood phylogeny (not a “sequence similarity network” as the authors suggest to call it) was estimated from an alignment of structurally conserved regions of the sequences. Given the absence of an outgroup, the tree was not possible to root.

While the structural similarity tree is consistent with the classification in steps 1 and 2, the phylogeny does not support what the authors claim. This is not because the tree is unrooted, but because there is no possible way to root it and claim that “the branching in the tree clearly reflects the four different groups of NATs” (lines 321-2), “Group 1 and Group 2 are closely related according to the tree” (lines 323-4) and “these enzymes [group 1] are obviously closely related” (lines 528-9) (among other similar claims). The orange and purple parts of group 1 can’t share a common ancestor that makes the two any more closely related to each other than to some other branches in the tree. What I see is that there is possible phylogenetic evidence for all the other groups – i.e. under reasonable rooting scenarios there’s monophyly for the groups – but not for group 1. There are different possible evolutionary explanations for this: early divergence from an ancestor with the group 1 motif, combined with divergence in other descendants is one, convergence is another. (The suggested archaeal common ancestor to NAA10 and NAA50 would fit well, and the tree suggests inclusion of NAA20, NAA60 and, possibly, NAA30 as descendants of an archaeal gene.)

Let me be clear that I’m not against presenting a classification based solely on steps 1) and 2), followed by an analysis based on phylogeny that investigates possible scenarios for the evolution of the proposed classes and groups, but there are claims in the text that I do not find supported by the data. An alternative is, as I suggested in my first review, to revise group 1 based on the phylogeny into two: NAA10 and NAA20 on the one hand and NAA30 on the other. Since NAA30 is not connected to any other cluster in the group in the SSN (fig. 2), separating it out does not seem to invalidate the principles behind the classification.

I’m glad the authors performed a separate analysis to show what an analysis of bacterial and archaeal sequences might look like. Extending the analysis to all three domains is, if at all possible, clearly an overwhelming task.

I fully accept the authors choice to keep the structure of the manuscript as a combined results and discussion manuscript. However, I think it would be fairer to call the current results section “Results and discussion” as that makes it clear that what follows is not just the authors’ own results but quite substantial discussion (evidenced also by the number of references). Given the length of the manuscript, ending with a Conclusions section would be helpful. This might mean moving some of what is in the current Discussion section to the R&D and some other material to the new Conclusions section. (For my taste, I think the M&M contains too much results, but I’m willing to turn a blind eye to this given the current length of the Results section.)

The manuscript is very long. Besides reworking the manuscript as suggested above (much of what is in the current discussion could be worked into already started arguments in the current results section), shortening should be possible in places; subsection 5.1 strikes me as a good candidate.

There are quite a number of typos and mistakes in the manuscript. I comment below on the ones I’ve found, but there are likely more of them.

Detailed comments:

Line 175: “closely related” should probably be “very similar”.

Line 208: The cluster containing NAA30 is shown in the figure.

Line 210: “NATs” should be “NAT”

Lines 258-9: You only specify what black and red mean.

Lines 282-3: See discussion above.

Figure 4: This tree is also inherently unrooted, but drawn as a rooted tree. Moreover, it’s drawn as a cladogram so that branch lengths are not informative. In order to judge anything about closeness, you should at least change so that the tree is drawn with proper branch lengths, i.e. branch lengths that reflect the structural differences you used to generate the tree.

Line 317: I think you should say here that the tree is not rooted because you don’t have an outgroup.

Lines 323, 326-7, 396: See discussion above

Figure 5: Please add the new, non-NAT, colours to the legend.

Lines 398-9: Delete “, and for which we cannot infer functions based on our computational

approach”. (You do make attempts to form hypothesizes about sequences that are in clusters without any characterized enzymes.)

Line 407: “NATs” should be “NAT”

Line 510: I’m not convinced that “topology” is the best word here, it leads my thoughts to protein structure.

Lines 579-80: What do you mean with “This observation indicates different evolutionary

paths for NATs, and not divergent evolution”?

Line 694: Do you have a reference for saying that 70% “usually indicates shared function”? In which protein family? (I have certainly seen proteins with much lower degree of identity that have the same function.)

Line 700: “sequences clustering together were closely related”: Perhaps I’m overly picky now, but you’re talking about sequence similarity, not evolutionary relationships.

Line 717: “as” should be “at”.

Line 731: “using algorithm” should be “using the algorithm”.

Line 749: “NAT” should be “NATs”.

Lines 753 and 755: “is” should be “was”.

Lines 756-7: Delete “The resulting dendrogram is simply a list of structural neighbors ranked by Dali Z-score.” It’s not a correct description of a distance-based phylogenetic tree (definitely not a “list”).

Lines 775-6: “MAFFT” is incorrectly spelled in two places.

Lines 792-3: Delete “The tree was not rooted.” Phylogenies that come out of programs are never rooted; you root them e.g. by including an outgroup.

Reviewer #3: The authors addressed all my concerns and suggestions in a satisfactory manner. The manuscript clearly improved and now is suitable for publication.

**Have all data underlying the figures and results presented in the manuscript been provided?**

Reviewer #1: Yes

Reviewer #3: Yes

PLOS authors have the option to publish the peer review history of their article (what does this mean?). If published, this will include your full peer review and any attached files.

Reviewer #1: No

Reviewer #3: No
---

## [Decision Letter · Decision Letter 2]

16 Oct 2020

Dear Prof. Reuter,

We are pleased to inform you that your manuscript 'Classification and phylogeny for the annotation of novel eukaryotic GNAT acetyltransferases' has been provisionally accepted for publication in PLOS Computational Biology.

Best regards,

Christos A. Ouzounis

Associate Editor

PLOS Computational Biology

Arne Elofsson

Deputy Editor

PLOS Computational Biology

Reviewer's Responses to Questions

**Comments to the Authors:**

Reviewer #1: Thank you for taking my suggestions into consideration. I have no further comments.

**Have all data underlying the figures and results presented in the manuscript been provided?**

Reviewer #1: Yes

PLOS authors have the option to publish the peer review history of their article (what does this mean?). If published, this will include your full peer review and any attached files.

Reviewer #1: No

---

## [Editor Report · Acceptance letter]

9 Dec 2020

PCOMPBIOL-D-20-00852R2 

Classification and phylogeny for the annotation of novel eukaryotic GNAT acetyltransferases

Dear Dr Reuter,

I am pleased to inform you that your manuscript has been formally accepted for publication in PLOS Computational Biology. Your manuscript is now with our production department and you will be notified of the publication date in due course.

With kind regards,

Nicola Davies
